# Make You Better: Reinforcement Learning from Human Gain

## Abstract

In human-agent collaboration tasks, it is essential to explore ways for developing assistive agents that can improve humans' performance in achieving their goals. In this paper, we propose the Reinforcement Learning from Human Gain (RLHG) approach, designed to effectively enhance human goal-achievement abilities in collaborative tasks with known human goals. Firstly, the RLHG method trains a value network to estimate primitive human performance in achieving goals. Subsequently, the RLHG method trains a gain network to estimate the positive gain of human performance in achieving goals when subjected to effective enhancement, in comparison to the primitive performance. The positive gains are used for guiding the agent to learn effective enhancement behaviors. Distinct from directly integrating human goal rewards into optimization objectives, the RLHG method largely mitigates the human-agent credit assignment issues encountered by agents in learning to enhance humans. We evaluate the RLHG agent in the widely popular Multi-player Online Battle Arena (MOBA) game, *Honor of Kings*, by conducting experiments in both simulated environments and real-world human-agent tests. Experimental results demonstrate that the RLHG agent effectively improves the goal-achievement performance of participants across varying levels.

## 1 Introduction

An intriguing research direction in the field of Artificial Intelligence (AI), particularly in the human-agent field, is how to effectively enhance human goal-achievement abilities within collaborative tasks. Human-Agent Collaboration (HAC) (Crandall *et al.*, 2018; Dafoe *et al.*, 2020) has gained significant attention from researchers, and numerous agents have been successfully developed to collaborate with humans in complex environments (Jaderberg *et al.*, 2019; Carroll *et al.*, 2019; Hu *et al.*, 2020; Strouse *et al.*, 2021; Bakhtin *et al.*, 2022; Gao *et al.*, 2023). However, as Amodei *et al.* (2016) stated, "[F]or an agent operating in a large, multifaceted environment, an objective function that focuses on only one aspect of the environment may implicitly express indifference over other aspects of the environment". The current agents focus mainly on maximizing their own rewards to complete the task, less considering the role of their human partners, which potentially leads to behaviors that are inconsistent with human preferences (Fisac *et al.*, 2020; Alizadeh Alamdari *et al.*, 2022). For instance, consider the scenario depicted in Figure 1, where there is an agent and a human on either side of an obstacle. Only the agent is capable of pushing or pulling the obstacle once. Both the human and the agent share the same task goal, i.e., obtaining the coin, while the human needs the agent's assistance to get the coin. In this scenario, the HAC agent may push the obstacle to the human side and pass through to get the coin by itself. However, in a qualitative study (Cerny, 2015) on companion behavior, humans reported greater enjoyment of the game when AI assisted them more like a sidekick. Thus, the human may prefer that the agent plays a more assisting role by pulling the obstacle to its side, thereby enabling the human to get the coin. To advance AI techniques for the betterment of humanity, it is crucial to consider ways to assist humans in improving their goal-achievement abilities rather than replacing them outright (Wilson and Daugherty, 2018).

Submitted to 37th Conference on Neural Information Processing Systems (NeurIPS 2023). Do not distribute.



Figure 1: Toy scenario, where an agent and a human are on either side of an obstacle. Only the agent is capable of pushing or pulling the obstacle once. They share the same task goal of obtaining the coin. ⇐: The agent replaces the human to get the coin by itself. ⇒: The agent assists the human to get the coin.

In complex collaborative environments, such as Multi-player Online Battle Arena (MOBA) games (Silva and Chaimowicz, 2017), humans pursue multiple individual goals, such as achieving higher MVP scores and experiencing more highlight moments, beyond simply winning the game to enhance their gaming experience (see Figure 4 (c), our participant survey). When human goals are aware, an intuitive approach to learning assistive agents would be to combine the agents' original rewards with the human's goal rewards (Hadfield-Menell *et al.*, 2016; Najar and Chetouani, 2021; Alizadeh Alamdari *et al.*, 2022). Nevertheless, directly incorporating the human's goal rewards may cause negative consequences, such as human-agent credit assignment issues, i.e., human rewards for achieving goals are assigned to non-assisting agents, which potentially leads the agent to learn poor behaviors and forfeits its autonomy. When human goals are unknown, some studies attempt to infer them from prior human behaviors using Bayesian Inference (BI) (Baker *et al.*, 2005; Foerster *et al.*, 2019; Puig *et al.*, 2020; Wu *et al.*, 2021) and Inverse Reinforcement Learning (IRL) (Ng *et al.*, 2000; Ziebart *et al.*, 2008; Ho and Ermon, 2016). Other work introduces auxiliary rewards, such as the human *empowerment* (Du *et al.*, 2020), i.e., the mutual information of human trajectories and current state, for guiding agents to learn assistive behaviors. However, the diverse and noisy human behaviors(Majumdar *et al.*, 2017) may be unrelated to actual human goals, leading agents to learn assistance behaviors that are not aligned with human preferences. Moreover, in tasks where human goals are known, these methods may not be as effective as explicitly modeling human goals (Du *et al.*, 2020; Alizadeh Alamdari *et al.*, 2022).

This paper focuses on the setting of known human goals in complex collaborative environments. Our key insight is that agents can enhance human goal-achievement abilities without compromising AI autonomy by learning from the human positive gains toward achieving goals under the agent's effective enhancement. We propose the Reinforcement Learning from Human Gain (RLHG) method, which aims to fine-tune a given pre-trained agent to be assistive in enhancing a given human model's performance in achieving specified goals. Specifically, the RLHG method involves two steps. Firstly, we determine the primitive performance of the human model in achieving goals. We train a value network to estimate the primitive human return in achieving goals with episodes collected by directly teaming the agent and the human to execute. Secondly, we train the agent to learn effective human enhancement behaviors. We train a gain network to estimate the positive gain of human return in achieving goals when subjected to effective enhancement, in comparison to the primitive performance. The agent is fine-tuned using the combination of its original advantage and the human-enhanced advantage calculated by the positive gains. The RLHG method can be seen as a plug-in that can be directly utilized to fine-tune any pre-trained agent to be assistive in human enhancement.

We conducted experiments in *Honor of Kings* (Wei *et al.*, 2022), one of the most popular MOBA games globally, which has received much attention from researchers lately (Ye *et al.*, 2020a,b,c; Gao *et al.*, 2021, 2023). We first evaluated the RLHG method in simulated environments, i.e., human model-agent tests. Our experimental results indicate that the RLHG agent is more effective than baseline agents in improving the human model goal-achievement performance. We further conducted real-world human-agent tests to verify the effectiveness of the RLHG agent. We tested the RLHG agent teaming up with different levels of participants. Our experimental results demonstrate that the RLHG agent could effectively improve the performance of general-level participants in achieving their individual goals to be close to those of high-level participants and that this enhancement can be generalized to different levels of participants. In general, our contributions are as follows:

- We propose a novel insight to effectively enhance human abilities in achieving goals within collaborative tasks by training an assistive agent to learn from human positive gains.

- We achieve our insight by proposing the RLHG algorithm and providing a practical implementation.

- We validated the effectiveness of the RLHG method by conducting human-agent tests in the complex MOBA game *Honor of Kings*.

## 2 Problem Settings

### 2.1 Game Introduction

MOBA games, characterized by multi-agent cooperation and competition mechanisms, long time horizons, enormous state-action spaces ($10^{20000}$), and imperfect information (OpenAI *et al.*, 2019; Ye *et al.*, 2020a), have attracted much attention from researchers. *Honor of Kings* is a renowned MOBA game played by two opposing teams on the same symmetrical map, each comprising five players. The game environment depicted in Figure 2 comprises the main hero with peculiar skill mechanisms and attributes, controlled by each player. The player can maneuver the hero's movement using the bottom-left wheel (C.1) and release the hero's skills through the bottom-right buttons (C.2, C.3). The player can view the local environment on the screen, the global environment on the top-left mini-map (A), and access game states on the top-right dashboard (B). Players of each camp compete for resources through team confrontation and collaboration, etc., with the task goal of winning the game by destroying the opposing team's crystal.

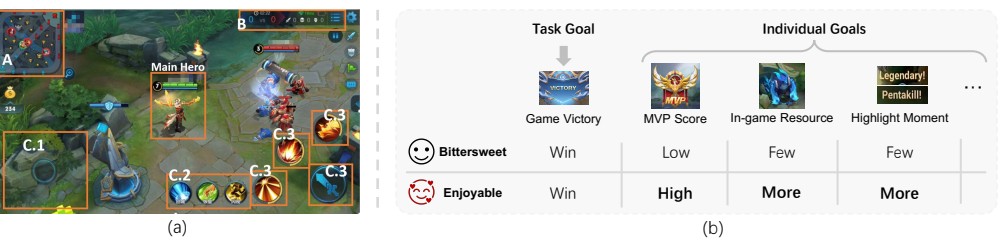

Figure 2: **(a)** The UI of *Honor of Kings*. **(b)** The player's goals in-game (based on our participant survey).

### 2.2 Human-Agent Enhancement

We formulate the human enhancement problem in collaborative tasks as an extension of the Dec-POMDP, which can be represented as a tuple $< N, \mathbf{S}, \mathbf{A}, \mathbf{O}, P, R, \gamma, \pi^H, \mathcal{G}^H, R^H >$, where $N$ denotes the number of agents. $\mathbf{S}$ denotes the space of global states. $\mathbf{A} = \{A_i, A^H\}_{i=1,...,N}$ denotes the space of actions of $N$ agents and a human to be enhanced, respectively. $\mathbf{O} = \{O_i, O^H\}_{i=1,...,N}$ denotes the space of observations of $N$ agents and the human, respectively. $P : \mathbf{S} \times \mathbf{A} \to \mathbf{S}$ and $R : \mathbf{S} \times \mathbf{A} \to \mathbb{R}$ denote the shared state transition probability function and reward function of $N$ agents, respectively. $\gamma \in [0, 1)$ denotes the discount factor. $\pi^H(a^H|o^H)$ is the human policy, which cannot be directly accessible to the agent. $\mathcal{G}^H = \{g_i\}_{i=1,...,M}$ denotes the human individual goals, where $g_i$ is a designated goal and $M$ is the total number of individual goals. $R^H : S \times \mathbf{A} \times \mathcal{G}^H \to \mathbb{R}$ denotes the goal reward function of the human. In agent-only scenarios, the optimization objective is to maximize the expected return $V^{\pi_\theta} = \mathbb{E}_{\pi_\theta}[G]$, where $G = \sum_{t=0}^{\infty} \gamma^t R_t$ is the discounted total rewards (OpenAI *et al.*, 2019; Ye *et al.*, 2020a). In human non-enhancement scenarios, the optimization objective is $V^{\pi_\theta, \pi^H} = \mathbb{E}_{\pi_\theta, \pi^H}[G] = \sum_a \pi_\theta(a|o, \pi^H)\mathbb{E}_{\pi^H}[G]$ (Carroll *et al.*, 2019; Strouse *et al.*, 2021). However, in human enhancement scenarios, the agent learns to enhance the human in achieving their goals $\mathcal{G}^H$. Therefore, the optimization objective can be formulated as:

$$V_{he}^{\pi_\theta, \pi^H} = V^{\pi_\theta, \pi^H} + \alpha \cdot V_H^{\pi_\theta, \pi^H} = \mathbb{E}_{\pi_\theta, \pi^H}[G + \alpha \cdot G_H] = \sum_a \pi_\theta(a|o, \pi^H)\mathbb{E}_{\pi^H}[G + \alpha \cdot G_H],$$

where $V_H^{\pi_\theta, \pi^H} = \mathbb{E}_{\pi_\theta, \pi^H}[G_H]$, $G_H = \sum_{t=0}^{\infty} \gamma^t R_t^H$ is the discounted total human goal rewards, and $\alpha$ is a balancing parameter. The agent's policy gradient can be formulated as:

$$g(\theta) = \nabla_\theta \log \pi_\theta(a|o, \pi^H)\mathbb{E}_{\pi^H}[A + \alpha \cdot A_H], \tag{1}$$

where $A = G - V^{\pi_\theta, \pi^H}$ and $A_H = G_H - V_H^{\pi_\theta, \pi^H}$ are the agent's original advantage and the human's enhanced advantage, respectively.

However, incorporating human rewards directly into the optimization objective may lead to negative consequences, such as human-agent credit assignment issues. Intrinsically, humans possess the primitive ability to achieve certain goals independently. Therefore, it is unnecessary to reward the agent for assisting in goals that the human can easily achieve, as it potentially impacts the agent's original behavior, resulting in losing its autonomy. In the subsequent section, we propose a novel insight to achieve effective human enhancement by instead learning from the positive gains that the human achieves goals better than his/her primitive performance.

## 3  Reinforcement Learning from Human Gain

In this section, We present the RLHG method in detail. We start with describing the key insight in the RLHG method (Section 3.1). Then we implement our insights and present the RLHG algorithm (Section 3.2). We end by providing a practical implementation of the RLHG algorithm (Section 3.3).

### 3.1  Effective Human Enhancement

In the process of learning to enhance humans, agents explore three types of behaviors: effective enhancement, invalid enhancement, and negative enhancement. Intuitively, effective enhancement can help humans achieve their goals better than their primitive performance, invalid enhancement provides no benefits for humans in achieving their goals but also causes no negative impact, and negative enhancement hinders humans from achieving their goals. Our key insight is that agents are only encouraged to learn effective enhancement behaviors, which we refer to learn from *positive gains*. Formally, we denote the effective enhancement policy as $\pi_\theta^{ef}$, the invalid enhancement policy as $\pi_\theta^{in}$, and the negative enhancement policy as $\pi_\theta^{ne}$. The agent's policy can be expressed as follows:

$$
\pi_\theta = \begin{cases} \pi_\theta^{ef}, & \text{if} \quad V_H^{\pi_\theta,\pi^H} > V_H^{\pi,\pi^H} \\ \pi_\theta^{in}, & \text{if} \quad V_H^{\pi_\theta,\pi^H} = V_H^{\pi,\pi^H} \\ \pi_\theta^{ne}, & \text{if} \quad V_H^{\pi_\theta,\pi^H} < V_H^{\pi,\pi^H} \end{cases} \tag{2}
$$

where $\pi$ is a given pre-trained policy and $V_H^{\pi,\pi^H}$ is the primitive value of the human policy $\pi^H$ teaming with $\pi$ to achieve goals. We use the $\rho$-function to represent the probability of exploring each policy, and we have $\rho(\pi_\theta^{ef}) + \rho(\pi_\theta^{in}) + \rho(\pi_\theta^{ne}) = 1$. Intuitively, the expected return of human goal-achievement under arbitrary enhancement is a lower bound of the expected return under effective enhancement, that is,

$$
V_H^{\pi_\theta^{ef},\pi^H} \geq \rho(\pi_\theta^{ef}) \cdot V_H^{\pi_\theta^{ef},\pi^H} + \rho(\pi_\theta^{in}) \cdot V_H^{\pi_\theta^{in},\pi^H} + \rho(\pi_\theta^{ne}) \cdot V_H^{\pi_\theta^{ne},\pi^H} = V_H^{\pi_\theta,\pi^H}.
$$

To ensure that the agent only learns effective enhancement behaviors, we replace the lower bound $V_H^{\pi_\theta,\pi^H}$ with $V_H^{\pi,\pi^H}$. Therefore, the agent's policy gradient 1 can be reformulated as:

$$
g(\theta) = \nabla_\theta \log \pi_\theta(a|o,\pi^H) \mathbb{E}_{\pi^H} \left[ A + \alpha \cdot \widehat{A}_H \right], \tag{3}
$$

where $\widehat{A}_H = (G_H - V_H^{\pi,\pi^H}) - \text{Gain}^{\pi_\theta^{ef},\pi^H}$ and $\text{Gain}^{\pi_\theta^{ef},\pi^H} = V_H^{\pi_\theta^{ef},\pi^H} - V_H^{\pi,\pi^H}$ is the expected of the effective enhancement benefit. We use $\text{Gain}_\omega$ to denote an estimate of $\text{Gain}^{\pi_\theta^{ef},\pi^H}$, which can be trained by minimizing the following loss function:

$$
L(\omega) = \mathbb{E}_{s \in S} \left[ I(G_H, V_\phi(s)) \cdot \| (G_H - V_\phi(s)) - \text{Gain}_\omega(s) \|_2 \right], \quad I(G,V) = \begin{cases} 1, & G > V \\ 0, & G \leq V \end{cases} \tag{4}
$$

where $I$ is an indicator function to filter invalid and negative enhancement samples and $V_\phi$ is an estimate of $V_H^{\pi,\pi^H}$.

### 3.2  The Algorithm

We achieve our insights and propose the RLHG algorithm as shown in Algorithm 1, which consists of two steps: the Human Primitive Value Estimation step and the Human Enhancement Training step.

**Human Primitive Value Estimation:** The RLHG algorithm initializes a value network $V_\phi(s)$, which is used to estimate the expected primitive human return for achieving $\mathcal{G}^H$ in state $s$. $V_\phi(s)$ is trained by minimizing the Temporal Difference (TD) errors (Sutton and Barto, 2018) with trajectory samples collected by teaming the agent $\pi$ and the human $\pi^H$ to execute in a collaboration environment. Afterward, $V_\phi(s)$ is frozen for subsequent human enhancement training.

**Human Enhancement Training:** The RLHG algorithm initializes the agent's policy network $\pi_\theta$ and value network $V_\psi$ by conditioned on the human policy $\pi^H$, respectively. The RLHG algorithm also initializes a value network $\text{Gain}_\omega(s)$, which is used to estimate the benefit value of the human return $G_H$ in state $s$ under effective enhancement over $V_\phi(s)$. $\text{Gain}_\omega(s)$ is trained by minimizing the loss function Eq. 4. The trajectory samples are also collected by teaming $\pi_\theta$ and $\pi^H$ to execute in

the collaboration environment. The agent's policy network $\pi_\theta$ is fine-tuned by the PPO (Schulman *et al.*, 2017) algorithm using the combination of the original advantage $A$ and the human-enhanced advantage $\widehat{A}_H$. The agent's value network $V_\psi$ is fine-tuned using the agent's original return $G$.

---

**Algorithm 1** Reinforcement Learning from Human Gain (RLHG)

---

**Require**: Human policy network $\pi^H$, human individual goals $\mathcal{G}^H$, agent policy network $\pi$, agent value network $V$, hyper-parameter $\alpha$
**Process**:

1: Initialize human primitive value network $V_\phi$;
   // Step I: Human Primitive Value Estimation
2: **while** not converged **do**
3:     Collect human-agent team $< \pi, \pi^H >$ trajectories;
4:     Compute human return $G_H$ for achieving goals $\mathcal{G}^H$;
5:     Update $V_\phi(s) \leftarrow G_H$
6: **end while**
7: Initialize agent policy network $\pi_\theta(a|o, \pi^H) \leftarrow \pi$, agent value network $V_\psi(s, \pi^H) \leftarrow V$, human gain network $\text{Gain}_\omega(s)$;
   // Step II: Human Enhancement Training
8: **while** not converged **do**
9:     Collect human-agent team $< \pi_\theta, \pi^H >$ trajectories;
10:     Compute agent original return $G$ and human return $G_H$;
11:     Compute agent original advantage $A = G - V_\psi(s, \pi^H)$;
12:     Compute human-enhanced advantage $\widehat{A}_H = (G_H - V_\phi(s)) - \text{Gain}_\omega(s)$;
13:     Update agent policy network $\pi_\theta \leftarrow A + \alpha \cdot \widehat{A}_H$;
14:     Update agent value network $V_\psi(s, \pi^H) \leftarrow G$;
15:     Update human gain network $\text{Gain}_\omega(s)$ with Eq. 4
16: **end while**

---

## 3.3 Practical Implementation

We provide the overall training framework of the RLHG algorithm, as shown in Figure 3. We elaborate on the integral components of the RLHG framework, including the human model, the agent model, and the training details.

**Human Model:** The RLHG algorithm introduces a human model as a partner of the agent during the training process. The human model can be trained via Behavior Cloning (BC) (Bain and Sammut, 1995) or any Supervised Learning (SL) techniques (Ye *et al.*, 2020b), but this is not the focus of our concern. The RLHG algorithm aims to fine-tune a pre-trained agent to enhance a given human model.

**Agent Model:** Any pre-trained agent can be used within our framework. Since in many practical scenarios agents cannot directly access human policies, we input the observed human historical info $h_t = (s_{t-m}^H, ..., s_t^H)$ into an LSTM (Hochreiter and Schmidhuber, 1997) module to extract the human policy embedding, similar to Theory-of-Mind (ToM) (Rabinowitz *et al.*, 2018). The human policy embedding is fed into two extra value networks, i.e., $V_\phi$ and $\text{Gain}_\omega$, and fused into the agent's original network. We use *surgery* techniques (Chen *et al.*, 2015; OpenAI *et al.*, 2019) to fuse the human policy embedding into the agent's original network, i.e. adding more randomly initialized units to an internal fully-connected layer. $V_\phi(h_t)$ and $\text{Gain}_\omega(h_t)$ output values estimate the human return for achieving goals without enhancement and the benefit under enhancement in state $s_t$, respectively.

**Training Details:** The overall training framework of the RLHG algorithm is shown in Figure 3. Figure 3 (a) shows the training process of the human primitive value network $V_\phi$, in which the agent's policy network is frozen. $V_\phi$ is trained by minimizing the TD errors. Figure 3 (b) shows the human enhancement training process, in which $V_\phi$ is frozen. The agent's policy and value networks are trained using the PPO algorithm. $\text{Gain}_\omega(h_t)$ is trained by minimizing the loss function Eq. 4. we apply the absolute activation function to ensure that the gains are non-negative. In practical training, we found that only conducting human enhancement training has a certain negative impact on the agent's original ability to complete the task. Therefore, we introduce $1 - \beta\%$ agent-only environment to maintain the agent's original ability and reserve $\beta\%$ human-agent environment to learn effective enhancement behaviors. These two environments can be easily controlled through the task gate, i.e., the task gate is set to 1 in the human-agent environment and 0 otherwise.

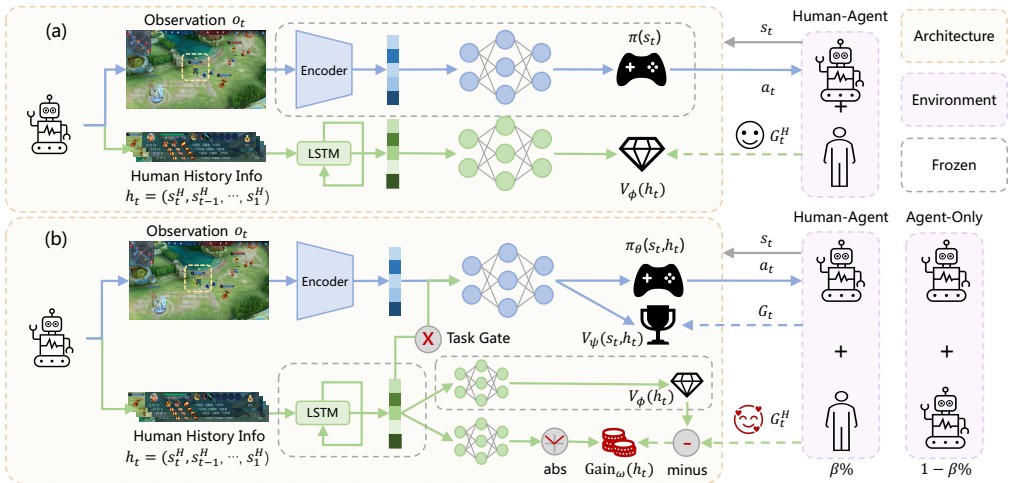

Figure 3: **The RLHG training framework.** (a) The human primitive value network $V_\phi$ is trained in the human-agent environment with the agent's policy $\pi$ frozen. (b) The human enhancement training framework. $V_\phi$ is frozen. $\beta\%$ human-agent environment is used to learn human enhancement behaviors, and $1 - \beta\%$ agent-only environment is used to maintain the agent's original ability.

## 4  Experiments

In this section, we evaluate the proposed RLHG method by conducting both simulated human model-agent tests and real-world human-agent tests in *Honor of Kings*. All experiments[1] were conducted in the 5v5 mode with a full hero pool (over 100 heroes, see Appendix A.2). Our demo videos and code are present at https://sites.google.com/view/rlhg-demo.

### 4.1  Experimental Setup

**Environment Setup:** To evaluate the performance of the RLHG agent, we conducted experiments in both the simulated environment, i.e., human model-agent game tests, and the real-world environment, i.e., human-agent game tests, as shown in Figure 4 (a) and (b), respectively. All game tests were played in a 5v5 mode, that is, 4 agents plus 1 human or human model team up against a fixed opponent team. To conduct our experiments, we communicated with the game provider and obtained testing authorization. The game provider assisted in recruiting 30 experienced participants with anonymized personal information, which comprised 15 high-level (top 1%) and 15 general-level (top30%) participants. We first did an IRB-approved participant survey on what top 5 goals participants want to achieve in-game, and the result is shown in Figure 4 (c). We can see that the top 5 goals voted the most by the 30 participants including the task goal, i.e., game victory, and 4 individual goals, i.e., high MVP score, high participation, more highlights, and more resources. We found that participants consistently rated the high MVP score individual goal most, even more than the task goal.

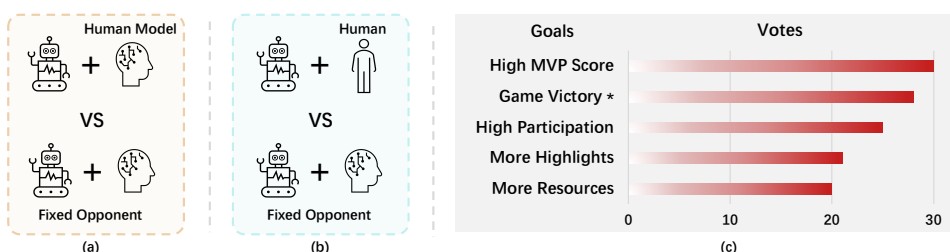

Figure 4: **Environment Setup. (a)** Simulated environment: the human model-agent game tests. **(b)** Real-world environment: the human-agent game tests. **(c)** Top 5 goals based on the stats of our participant survey. * denotes the task goal. The participant survey contains 8 initial goals, each participant can vote up to 5 non-repeating goals, and can also add additional goals. 30 participants voluntarily participated in the voting.

**Training Setup:** We were authorized to use the Wukong agent (Ye *et al.*, 2020a) as the pre-trained agent and use the JueWu-SL agent (Ye *et al.*, 2020b) as the fixed human model. Note that both

---

[1]All experiments are conducted subject to oversight by an Institutional Review Board (IRB).

the Wukong agent and the JueWu-SL agent were developed at the same level as the high-level (top 1%) players. We adopted the top 4 individual goals as $\mathcal{G}$ for the pre-trained agent to enhance the human model. The corresponding goal reward function can be found in Appendix B.3. We trained the human primitive value network and fine-tune the agent until they converge for 12 and 40 hours, respectively, using a physical computer cluster with 49600 CPU cores and 288 NVIDIA V100 GPU cards. The batch size of each GPU is set to 256. The hyper-parameters $\alpha$ and $\beta$ are set to 2 and 50, respectively. The step size and unit size of the LSTM module are set to 16 and 4096, respectively. Due to space constraints, detailed descriptions of the network structure and ablation studies on these hyper-parameters can be found in Appendix B.6 and Appendix C.1, respectively.

**Baseline Setup:** We compared the RLHG agent with two baseline agents: the Wukong agent (the pre-trained agent) and the Human Reward Enhancement (HRE) agent (the pre-trained agent learns to be assistive by incorporating the human's goal rewards). The human model-agent team (4 Wukong agents plus 1 human model) was adopted as the fixed opponent for all tests. For fair comparisons, both the HRE and RLHG agents are trained using the same goal reward function, and all common parameters and training resources are kept consistent. Results are reported over five random seeds.

## 4.2 Human Model-Agent Test

Directly evaluating agents with humans is expensive, which is not conducive to model selection and iteration. Instead, we build a simulated environment, i.e., human model-agent game tests, to evaluate agents, in which the human model, i.e., the JueWu-SL agent, teams up with different agents.

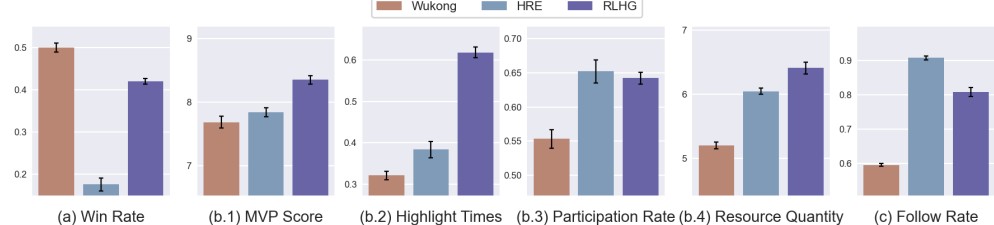

Figure 5: The performance of the human model in achieving game goals after teaming up with different agents. **(a)** The task goal. **(b)** The top 4 individual goals (b.1, b.2, b.3, and b.4). **(c)** The follow rate metric: the frequency with which an agent follows a human in the entire game. Each agent played 10,000 games. Error bars represent 95% confidence intervals, calculated over games.

Figure 5 shows the results of the human model on different game goals, including the top 4 individual goals and the task goal, i.e., the Win Rate, after teaming up with different agents. From Figure 5 (b), we can observe that both the RLHG agent and the HRE agent significantly enhance the performance of the human model in achieving the top 4 individual goals, and the RLHG agent has achieved the best enhancement effect on most of the individual goals. However, as shown in Figure 5 (a), the HRE agent drops significantly on the task goal. We observed the actual performance of the HRE agent teamed with the human model and found that the HRE agent did many unreasonable behaviors. For example, to assist the human model in achieving the goals of Participation Rate and Highlight Times, the HRE agent had been following the human model throughout the entire game, such excessive following behaviors greatly affect its original ability to complete the task and lead to a decreased Win Rate. This can also be reflected in Figure 5(c), in which the HRE agent has the highest Follow-Rate metric. Although the Follow-Rate of the RLHG agent has also increased, we observed that most of the following behaviors of the RLHG agent can effectively assist the human model. We also found that the Win Rate of the RLHG agent decreased slightly, which is in line with expectations because the RLHG agent made certain sacrifices to the task goal while enhancing humans in achieving their individual goals. In practical applications, we implemented an adaptive adjustment mechanism by simply utilizing the agent's original value network to measure the degree of completing the task goal and setting the task gate to 1 (enhancing the human) when the original value is above the specified threshold $\xi$, and to 0 (completing the task) otherwise. The threshold $\xi$ depends on the human preference, i.e. the relative importance of the task goal and the human's individual goals. We verify the effectiveness of the adaptive adjustment mechanism in Appendix C.2.

## 4.3 Human-Agent Test

In this section, we conduct online experiments to examine whether the RLHG agent can effectively enhance human participants (We did not compare the HRE agent, since the HRE agent learned lots

of unreasonable behaviors, resulting in a low Win Rate). We used a within-participant design for the experiment: each participant teams up with four agents. This design allowed us to evaluate both objective performances as well as subjective preferences. All participants read detailed guidelines and provided informed consent before the testing. Each participant tested 20 matches. After each test, participants reported their preference over their agent teammates. For fair comparisons, participants were not told the type of their agent teammates. See Appendix D for additional experimental details, including experimental design, result analysis, and ethical review.

Table 1: The results of **high-level** participants achieving goals after teaming up with different agents. Results for the task goal are expressed as mean, and results for individual goals are expressed as mean (std.).

| Agent \ Goals | Task Goal | Top 4 Individual Goals | | | |
| --- | --- | --- | --- | --- | --- |
| | Win Rate | MVP Score | Highlight Times | Participation Rate | Resource Quantity |
| Wukong | 52% | 8.86 (0.79) | 0.53 (0.21) | 0.46 (0.11) | 5.3 (2.87) |
| RLHG | 46.7% | **10.28** (0.75) | **0.87** (0.29) | **0.58** (0.09) | **6.28** (2.71) |

Table 2: The results of **general-level** participants achieving goals after teaming up with different agents. Results for the task goal are expressed as mean, and results for individual goals are expressed as mean (std.).

| Agent \ Goals | Task Goal | Top 4 Individual Goals | | | |
| --- | --- | --- | --- | --- | --- |
| | Win Rate | MVP Score | Highlight Times | Participation Rate | Resource Quantity |
| Wukong | 34% | 7.44 (0.71) | 0.37 (0.349) | 0.41 (0.11) | 4.98 (2.73) |
| RLHG | 30% | **9.1** (0.61) | **0.75** (0.253) | **0.59** (0.05) | **5.8** (2.78) |

We first compare the objective performance of the participants on different goal-achievement metrics after teaming up with different agents. Table 1 and Table 2 demonstrate the results of high-level and general-level participants, respectively. We see that both high-level and general-level participants had significantly improved their performance on all top 4 individual goals after teaming up with the RLHG agent. Notably, the RLHG agent effectively improves the performance of general-level participants in achieving individual goals even better than the primitive performance of high-level participants. We also notice that the Win Rate of the participants decreased when they teamed up with the RLHG agent, which is consistent with the results in the simulated environment. However, we find in the subsequent subjective preference statistics that the improvement of Gaming Experience brought by the enhancement outweighs the negative impact of the decrease in Win Rate.

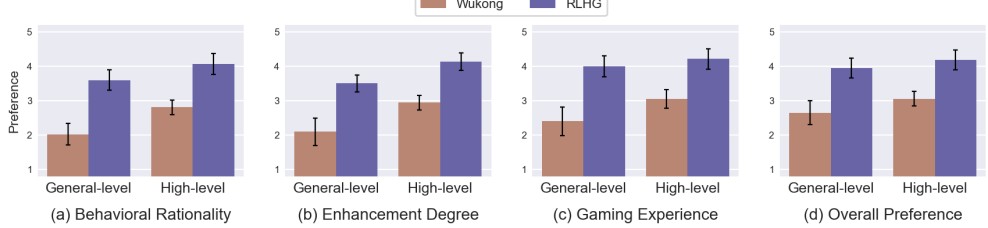

Figure 6: **Participants' preference over their agent teammates.** (a) Behavioral Rationality: the reasonableness of the agent's behavior. (b) Enhancement Degree: The degree to which the agent enhances your abilities to achieve your goals. (c) Gaming Experience: your overall gaming experience. (d) Overall Preference: your overall preference for your agent teammates. Participants scored (1: Terrible, 2: Poor, 3: Normal, 4: Good, 5: Perfect) in these metrics after each game test. Error bars represent 95% confidence intervals, calculated over games. See Appendix D.2.3 for detailed wording and scale descriptions.

We then compare the subjective preference metrics, i.e., the Behavioral Rationality, the Enhancement Degree, the Gaming Experience, and the Overall Preference, reported by participants over their agent teammates, as shown in Figure 6. We find that most participants showed great interest in the RLHG agent, and they believed that the RLHG agent's enhancement behaviors were more reasonable than that of the Wukong agent, and the RLHG agent's enhancement behaviors brought them a better gaming experience. A high-level participant commented on the RLHG agent "The agent frequently helps me do what I want to do, and this feeling is amazing." In general, participants were satisfied with the RLHG agent and gave higher scores in the Overall Preference metric (Figure 6 (d)).

### 4.4 Case Study

To better understand how the RLHG agent effectively enhances participants, we visualize the values predicted by the gain network in two test scenarios where participants benefitted from the RLHG agent's assistance, as illustrated in Figure 7. In the first scenario (Figure 7 (a)), the RLHG agent successfully assisted the participant in achieving the highlight goal, whereas the Wukong agent disregards the participant, leading to a failure in achieving the highlight goal. The visualization (Figure 7 (b)) of the gain network illustrates that the gain of the RLHG agent, when accompanying the participant, is positive, reaching the maximum when the participant achieved the highlight goal. In the second scenario (Figure 7 (c)), the RLHG agent actively relinquishes the acquisition of the monster resource, enabling the participant to successfully achieve the resource goal. Conversely, the Wukong agent competes with the participant for the monster resource, resulting in the participant's failure to achieve the resource goal. The visualization (Figure 7 (d)) of the gain network also reveals that the gain of the RLHG agent's behavior - actively forgoing the monster resource, is positive, with the gain peaking when the participant achieved the resource goal. These results indicate that the RLHG agent learns effective enhancement behaviors through the guidance of the gain network.

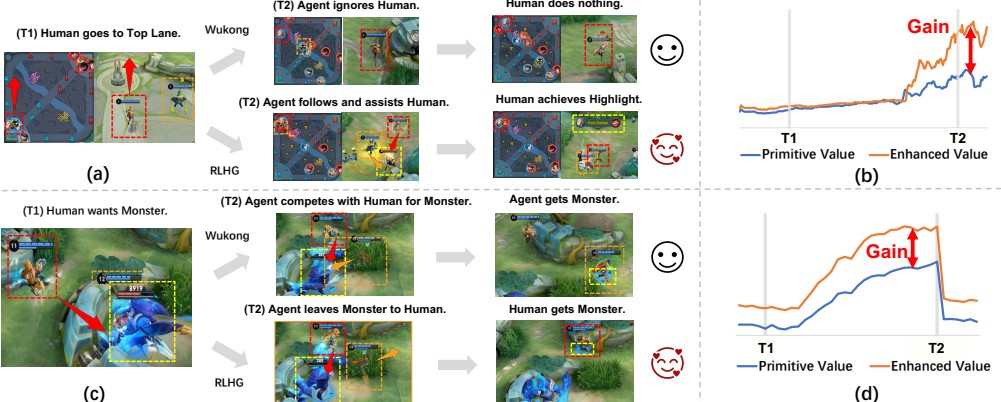

Figure 7: **The RLHG agent enhances participants in two scenarios. (a)** The Wukong agent ignores the participant; The RLHG agent accompanies the participant and assists the participant in achieving the highlight goal. **(b)** The gain value in scenario (a). **(c)** The Wukong agent competes with the participant for the monster resource; The RLHG agent actively forgoes the monster resource, and the participant successfully achieves the resource goal. **(d)** The gain value in scenario (c).

## 5 Discussion and Conclusion

**Summary.** In this work, we introduced the Reinforcement Learning from Human Gain method, dubbed RLHG, designed to effectively enhance human goal-achievement abilities within collaborative tasks. The RLHG method first trains a value network to estimate the primitive performance of humans in achieving goals. Subsequently, the RLHG method trains a gain network to estimate the positive gain of human performance in achieving goals under effective enhancement over that of the primitive. The positive gains are used for guiding the agent to learn effective enhancement behaviors. The RLHG method can be regarded as a continual learning plug-in that can be directly utilized to fine-tune any pre-trained agent to be assistive in human enhancement. The experimental results in *Honor of Kings* demonstrate that the RLHG agent effectively improves the performance of general-level participants in achieving their individual goals to be close to those of high-level participants and that this enhancement is generalizable across participants at different levels.

**Limitations and Future Work.** In this work, we only focus on the setting of known human goals. But for many practical complex applications, human goals may be difficult to define and formalize, and the goal reward function needs to be inferred using Inverse Reinforcement Learning (IRL) (Ng *et al.*, 2000; Ziebart *et al.*, 2008; Ho and Ermon, 2016) or Reinforcement Learning from Human Feedback (RLHF) (Christiano *et al.*, 2017; Ibarz *et al.*, 2018; Ouyang *et al.*, 2022) techniques. Future work can combine the RLHG method with goal inference methods to solve complex scenarios where human goals are unknown. Besides, our method and experiments only consider the scenario where multiple agents enhance one human. Another worthy research direction is how to simultaneously enhance multiple humans with diverse behaviors.

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
