# A Environment Details

## A.1 Game Introduction

MOBA (Multiplayer Online Battle Arena) games, characterized by multi-agent cooperation and competition mechanisms, long time horizons, enormous state-action spaces ($10^{20000}$), and imperfect information (OpenAI *et al.*, 2019; Ye *et al.*, 2020), have attracted much attention from researchers. *Honor of Kings* is a renowned MOBA game played by two opposing teams on the same symmetrical map, each comprising five players. The game environment depicted in Figure 1 comprises the main hero with peculiar skill mechanisms and attributes, controlled by each player. The player can maneuver the hero's movement using the bottom-left wheel (C.1) and release the hero's skills through the bottom-right buttons (C.2, C.3). The player can view the local environment on the screen, the global environment on the top-left mini-map (A), and access game stats on the top-right dashboard (B). Players of each camp compete for resources through team confrontation and collaboration, etc., with the task goal of winning the game by destroying the opposing team's crystal. The gaming experience is vital to a player's engagement and satisfaction. Along with the task goal, players also pursue multiple individual goals (see Appendix D.2.3), such as achieving a higher MVP score, experiencing more highlight moments, and obtaining more in-game resources, among others. The pursuit of these goals can contribute to a more enjoyable and rewarding gaming experience.

For fair comparisons, all experiments in this paper were carried out using a fixed released game engine version (Version 8.2 series) of *Honor of Kings*.

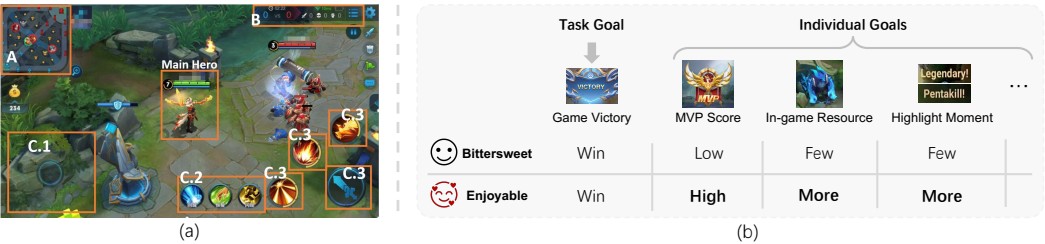

Figure 1: **(a)** The UI interface of *Honor of Kings*. **(b)** The player's goals in-game.
.

## A.2 Hero Pool

Table 1 shows the full hero pool used in Experiments. Each match involves two camps playing against each other, and each camp consists of five randomly picked heroes.

Table 1: Hero pool used in **Experiments**.

| | |
|---|---|
| Full Hero pool | Lian Po, Xiao Qiao, Zhao Yun, Mo Zi, Da Ji, Ying Zheng, Sun Shangxiang, Luban Qihao, Zhuang Zhou, Liu Chan Gao Jianli, A Ke, Zhong Wuyan, Sun Bin, Bian Que, Bai Qi, Mi Yue, Lv Bu, Zhou Yu, Yuan Ge, Chengji Sihan Xia Houdun, Zhen Ji, Cao Cao, Dian Wei, Gongben Wucang, Li Bai, Make Boluo, Di Renjie, Da Mo, Xiang Yu Wu Zetian, Si Mayi, Lao Fuzi, Guan Yu, Diao Chan, An Qila, Cheng Yaojin, Lu Na, Jiang Ziya, Liu Bang, Chang E Han Xin, Wang Zhaojun, Lan Lingwang, Hua Mulan, Ai Lin, Zhang Liang, Buzhi Huowu, Nake Lulu, Ju Youjing Ya Se, Sun Wukong, Niu Mo, Hou Yi, Liu Bei, Zhang Fei, Li Yuanfang, Yu Ji, Zhong Kui, Yang Yuhuan, Zhu Bajie Yang Jian, Nv Wa, Ne Zha, Ganjiang Moye, Ya Dianna, Cai Wenji, Taiyi Zhenren, Donghuang Taiyi, Gui Guzi Zhu Geliang, Da Qiao, Huang Zhong, Kai, Su Lie, Baili Xuance, Baili Shouyue, Yi Xing, Meng Qi, Gong Sunli Shen Mengxi, Ming Shiyin, Pei Qinhu, Kuang Tie, Mi Laidi, Yao, Yun Zhongjun, Li Xin, Jia Luo, Dun Shan, Sun Ce Shangguan Waner, Ma Chao, Dong Fangyao, Xi Shi, Meng Ya, Luban Dashi, Pan Gu, Meng Tian, Jing, A Guduo Xia Luote, Lan, Sikong Zhen, Erin, Yun ying, Jin Chan, Fei, Sang Qi, Ge Ya, Hai Yue, Zhao Huaizhen, Lai Xiao |

# B Framework Details

## B.1 Infrastructure Design

Figure 2 shows the infrastructure of the training system (Ye *et al.*, 2020), which consists of four key components: AI Server, Inference Server, RL Learner, and Memory Pool. The AI Server (the actor)

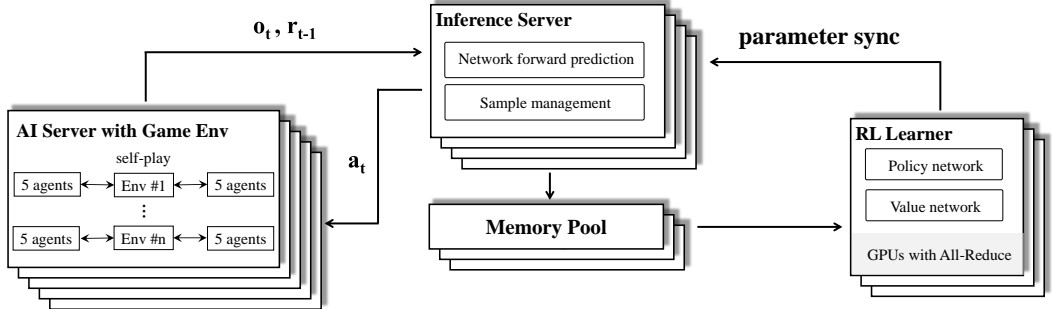

Figure 2: The infrastructure of the training system.

covers the interaction logic between the agents and the environment. The Inference Server is used for the centralized batch inference on the GPU side. The RL Learner (the learner) is a distributed training environment for RL models. And the Memory Pool is used for storing the experience, implemented as a memory-efficient circular queue.

## B.2 Task Reward Design

Table 2 demonstrates the details of the designed task reward from environment.

Table 2: The details of the environment reward.

| Head | Reward Item | Weight | Type | Description |
|------|-------------|--------|------|-------------|
| Farming Related | Gold | 0.005 | Dense | The gold gained. |
| | Experience | 0.001 | Dense | The experience gained. |
| | Mana | 0.05 | Dense | The rate of mana (to the fourth power). |
| | No-op | -0.00001 | Dense | Stop and do nothing. |
| | Attack monster | 0.1 | Sparse | Attack monster. |
| KDA Related | Kill | 1 | Sparse | Kill a enemy hero. |
| | Death | -1 | Sparse | Being killed. |
| | Assist | 1 | Sparse | Assists. |
| | Tyrant buff | 1 | Sparse | Get buff of killing tyrant, dark tyrant, storm tyrant. |
| | Overlord buff | 1.5 | Sparse | Get buff of killing the overlord. |
| | Expose invisible enemy | 0.3 | Sparse | Get visions of enemy heroes. |
| | Last hit | 0.2 | Sparse | Last hitting an enemy minion. |
| Damage Related | Health point | 3 | Dense | The health point of the hero (to the fourth power). |
| | Hurt to hero | 0.3 | Sparse | Attack enemy heroes. |
| Pushing Related | Attack turrets | 1 | Sparse | Attack turrets. |
| | Attack crystal | 1 | Sparse | Attack enemy home base. |
| Win/Lose Related | Destroy home base | 4 | Sparse | Destroy enemy home base. |

## B.3 Human Goal Reward Design

Table 3 demonstrates the details of the designed human goal reward.

## B.4 Feature Design

Table 4 shows the designed features of the Wukong agent (Ye et al., 2020), some of which (observable) are used as human features.

Table 3: The details of the human goal reward.

| Head | Reward Item | Weight | Type | Description |
|------|-------------|--------|------|-------------|
| MVP Score Related | Kill | 1 | Sparse | Kill a enemy hero. |
| | Death | -1 | Sparse | Being killed. |
| | Assist | 1 | Sparse | Assists. |
| | Hurt to hero | 0.3 | Sparse | Attack enemy heroes. |
| | Health point | 3 | Dense | The health point of the hero (to the fourth power). |
| Participation Related | Participation | 1 | Dense | Percentage of players participating in the team fight. |
| Highlight Related | Highlight | 2 | Sparse | Double kill, triple kill, quadra kill, penta kill. |
| Resource Related | Buff | 1 | Sparse | Get a red buff, blue buff. |
| | Health cake | 1 | Sparse | Get a health cake. |

Table 4: The observation space of agents. ∗ are used as human features.

| Feature Class | Field | Description | Dimension | Type |
|---------------|-------|-------------|-----------|------|
| **1. Unit feature** | Scalar | Includes heroes, minions, monsters, and turrets | 8599 | |
| Heroes* | Status | Current HP, mana, speed, level, gold, KDA, buff, bad states, orientation, visibility, etc. | 1842 | (one-hot, normalized float) |
| | Position | Current 2D coordinates | 20 | (normalized float) |
| | Attribute | Is main hero or not, hero ID, camp (team), job, physical attack and defense, magical attack and defense, etc. | 1330 | (one-hot, normalized float) |
| | Skills | Skill 1 to Skill N's cool down time, usability, level, range, buff effects, bad effects, etc. | 2095 | (one-hot, normalized float) |
| | Item | Current item lists | 60 | (one-hot) |
| Minions | Status | Current HP, speed, visibility, killing income, etc. | 1160 | (one-hot, normalized float) |
| | Position | Current 2D coordinates | 80 | (normalized float) |
| | Attribute | Camp (team) | 80 | (one-hot) |
| | Type | Type of minions (melee creep, ranged creep, siege creep, super creep, etc.) | 200 | (one-hot) |
| Monsters* | Status | Current HP, speed, visibility, killing income, etc. | 868 | (one-hot, normalized float) |
| | Position | Current 2D coordinates | 56 | (normalized float) |
| | Type | Type of monsters (normal, blue, red, tyrant, overlord, etc.) | 168 | (one-hot) |
| Turrets | Status | Current HP, locked targets, attack speed, etc. | 520 | (one-hot, normalized float) |
| | Position | Current 2D coordinates | 40 | (normalized float) |
| | Type | Type of turrets (tower, high tower, crystal, etc.) | 80 | (one-hot) |
| **2. In-game stats feature** | Scalar | Real-time statistics of the game | 68 | |
| Static statistics* | Time | Current game time | 5 | (one-hot) |
| | Gold | Golds of two camps | 12 | (normalized float) |
| | Alive heroes | Number of alive heroes of two camps | 10 | (one-hot) |
| | Kill | Kill number of each camp (Segment representation) | 6 | (one-hot) |
| | Alive turrets | Number of alive turrets of two camps | 8 | (one-hot) |
| Comparative statistics* | Gold diff | Gold difference between two camps (Segment representation) | 5 | (one-hot) |
| | Alive heroes diff | Alive heroes difference between two camps | 11 | (one-hot) |
| | Kill diff | Kill difference between two camps | 5 | (one-hot) |
| | Alive turrets diff | Alive turrets difference between two camps | 6 | (one-hot) |
| **3. Invisible opponent information** | Scalar | Invisible information used for the value net | 560 | |
| Opponent heroes | Position | Current 2D coordinates, distances, etc. | 120 | (normalized float) |
| NPC | Position | Current 2D coordinates of all non-player characters, including minions, monsters, and turrets | 440 | (normalized float) |
| **4. Spatial feature** | Spatial | 2D image-like, extracted in channels for convolution | 7x17x17 | |
| Skills* | Region | Potential damage regions of ally and enemy skills | 2x17x17 | |
| | Bullet* | Bullets of ally and enemy skills | 2x17x17 | |
| Obstacles* | Region | Forbidden region for heroes to move | 1x17x17 | |
| Bushes* | Region | Bush region for heroes to hide | 1x17x17 | |
| Health cake* | Region | Cake for heroes to recover blood | 1x17x17 | |

## B.5 Action Design

Table 5 shows the action space of agents.

Table 5: The action space of agents.

| Action | Detail | Description |
|---|---|---|
| What | Illegal action | Placeholder. |
| | None action | Executing nothing or stopping continuous action. |
| | Move | Moving to a certain direction determined by move x and move y. |
| | Normal Attack | Executing normal attack to an enemy unit. |
| | Skill1 | Executing the first skill. |
| | Skill2 | Executing the second skill. |
| | Skill3 | Executing the third skill. |
| | Skill4 | Executing the fourth skill (only a few heroes have Skill4). |
| | Summoner ability | An additional skill choosing before the game begins (10 to choose). |
| | Return home(Recall) | Returning to spring, should be continuously executed. |
| | Item skill | Some items can enable an additional skill to player's hero. |
| | Restore | Blood recovering continuously in 10s, can be disturbed. |
| | Collaborative skill | Skill given by special ally heroes. |
| How | Move X | The x-axis offset of moving direction. |
| | Move Y | The y-axis offset of moving direction. |
| | Skill X | The x-axis offset of a skill. |
| | Skill Y | The y-axis offset of a skill. |
| Who | Target unit | The game unit(s) chosen to attack. |

## B.6 Network Architecture

Figure 3 shows the detailed network architecture of the RLHG agent, which consists of two parts: the pre-trained Wukong model (Ye *et al.*, 2020), and the human enhancement module.

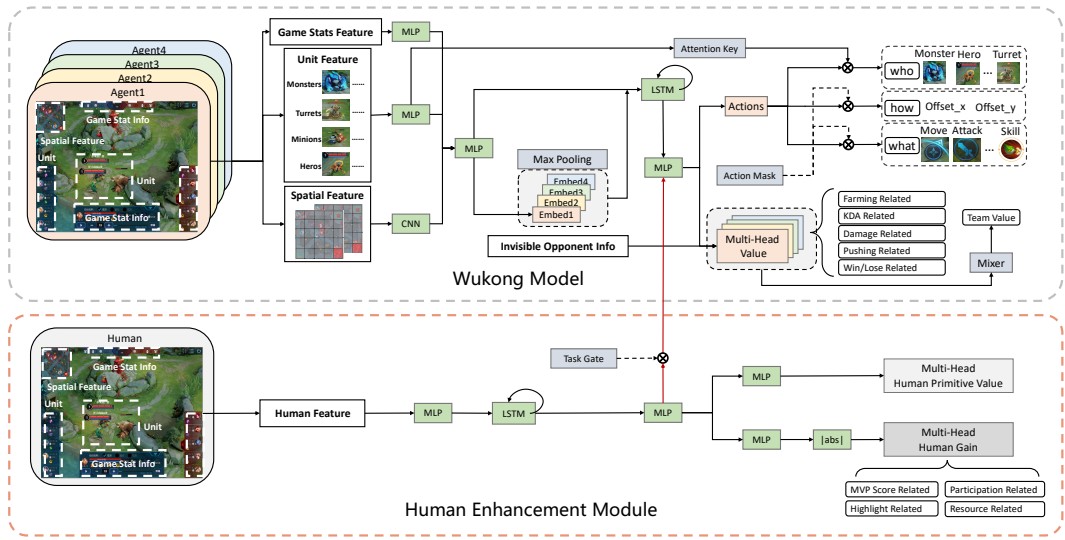

Figure 3: The network structure.

**Human Enhancement Module.** Human features are sequentially fed into the Fully-Connected (FC) layers with LSTM (Hochreiter and Schmidhuber, 1997) to extract human policy embedding. The policy embedding is used to predict human primitive values and gains. We apply the absolute activation function to ensure that the gains are non-negative. To manage the uncertain value of state-action in the game, we introduce the multi-head value estimation (Ye *et al.*, 2020) into the network by grouping the human goal reward in Table 3.

**Human Conditioned Policy Modeling.** We use surgery techniques (Chen *et al.*, 2015; OpenAI *et al.*, 2019) to fuse the human policy embedding into the agent's original network, i.e. adding more randomly initialized units to an internal FC layer. The task gate is used to control the agent's policy

style, i.e., for the non-enhancement mode, the task gate is set to 0, and for the enhancement mode, the task gate is set to 1. The agent's policy network predicts a sequence of actions for each agent based on its observation and human policy embedding.

**Network Parameter Details.** All hyper-parameters of the Wukong model are consistent with the original (Ye *et al.*, 2020). The unit size and step size of the LSTM module in the human enhancement module are set to 4096 and 16, respectively. The parameters of each FC layer are shown in our code. We use Adam (Kingma and Ba, 2014) with an initial learning rate of 0.0001 for fine-turning.

# C    Supplementary Experiment

## C.1    Ablation Study

We examine the influence of the balance parameter $\alpha$, i.e., the relative importance of human individual goals relative to the task goal. The results of RLHG agents trained with different values of $\alpha$ are shown in Figure 4. We can see that with the increase of $\alpha$, the human model's performance in achieving individual goals is significantly improved, but the negative effect is that the agent sacrifices its original ability to achieve the task goal (The Win Rate metric is reduced). We also notice that when $\alpha$ is too large, the Win Rate is significantly reduced, which will also have a negative impact on the MVP score goal. We find that when $\alpha$ is set to 2, it not only greatly improves the human model's performance in achieving individual goals, but also has little impact on the Win Rate. Therefore, in our experiments, $\alpha$ is set to 2.

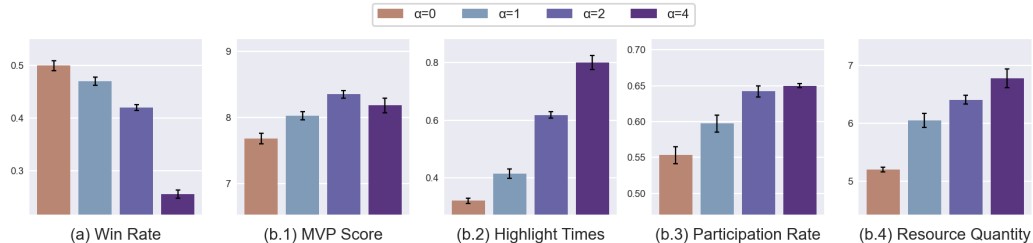

Figure 4: Influence of the balance parameter $\alpha$. Note that $\alpha = 0$ means training without enhancement.

## C.2    Adaptive Adjustment Mechanism

We implement an adaptive adjustment mechanism by simply utilizing the agent's original value network to measure the degree of completing the task goal. We first normalize the output of the original value network and then set the task gate to 1 (enhancing the human) when the normalized value is above the specified threshold $\xi$, and to 0 (completing the task) otherwise. The threshold $\xi$ is used to control the timing of enhancement. The results of RLHG agents with different values of $\xi$ are shown in Figure 5. We can see that as the threshold $\xi$ increases, the Win Rate increases, and the human model's performance on individual goals decreases. In practical applications, the threshold $\xi$ can be set according to human preference.

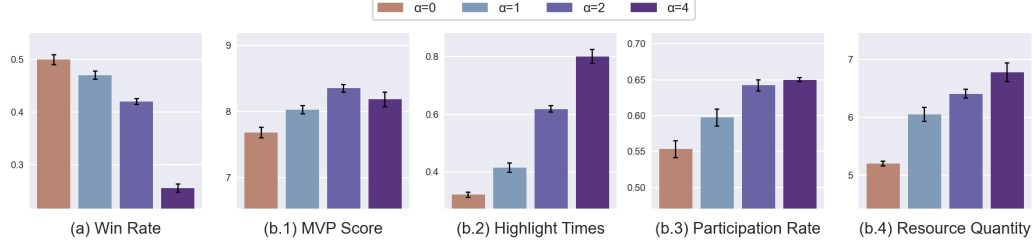

Figure 5: Influence of the threshold $\xi$. Note that $\xi = 1$ means never enhancement, and $\xi = 0$ means always enhancement.

# D    Details of Human-Agent Collaboration Test

## D.1    Ethical Review

The ethics committee of a third-party organization conducted an ethical review of our project. They reviewed our experimental procedures and risk avoidance methods (see Appendix D.1.3). They believed that our project complies with the "New Generation of AI Ethics Code" [1] of the country to which the participants belonged (China), so they approved our study. In addition, all participants consented to the experiment and provided informed consent (see Appendix D.1.1) for the study.

### D.1.1    Informed Consent

All participants were told the following experiment guidelines before testing:

- This experiment is to study human-agent collaboration technology in MOBA games.
- Your identity information will not be disclosed to anyone.
- All game statistics are only used for academic research.
- You will be invited into matches where your opponents and teammates are agents.
- Your goal is to win the game as much as possible by collaborating with agent teammates.
- Your agent teammates will assist you in achieving your individual goals in the game.
- After each test, you can report your preference over the agent teammates.
- After each test, you may also voluntarily fill out a debriefing questionnaire to tell us your open-ended feedback about the agent teammates.
- Each game lasts 10-20 minutes.
- You may voluntarily choose whether to take the test. You can terminate the test at any time if you feel unwell during the test.
- At any time, if you want to delete your data, you can contact the game provider directly to delete it.

If participants volunteer to take the test, they will first provide written informed consent, then we will provide them with the equipment and game account, and explain the experimental details on the screen.

### D.1.2    Screenshots

Screenshots of detailed experimental instructions are shown below.

1. Read tutorial and instruction on the study and gameplay. (Figure 6)
2. Read the detailed test content and precautions. (Figure 7)
3. Play the game with agents until the game is complete. (Figure 8)
4. Answer questions about perceptions and preferences.(Figure 9, 10, 11, and 12)
5. Volunteer to complete a debriefing questionnaire regarding open-ended feedback from your agent teammates. (Figure 13)
6. Repeat steps 3, 4, and 5 for a total of 20 times.

After the participant has read it carefully and confirmed complete understanding, the test will begin.

---

[1]China: MOST issues New Generation of AI Ethics Code, https://www.dataguidance.com/news/china-most-issues-new-generation-ai-ethics-code

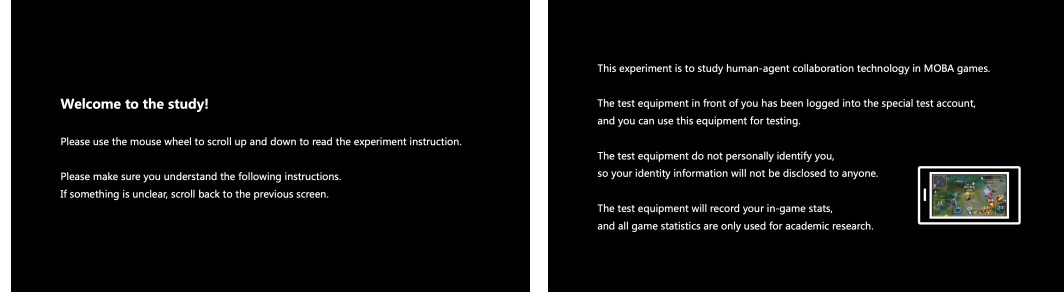

(a) Welcome participants to the experiment.      (b) Introduce test equipment.

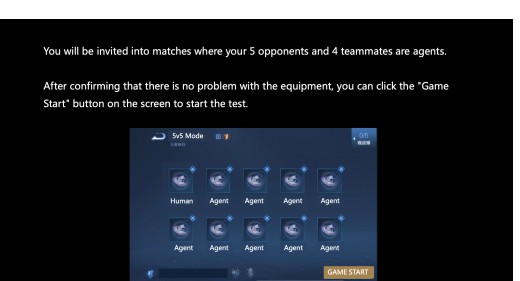
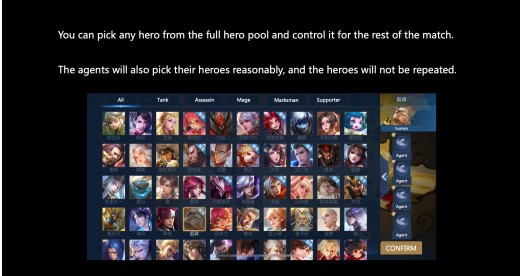

(c) Introduces test mode.      (d) Introduce participant's controllable hero.

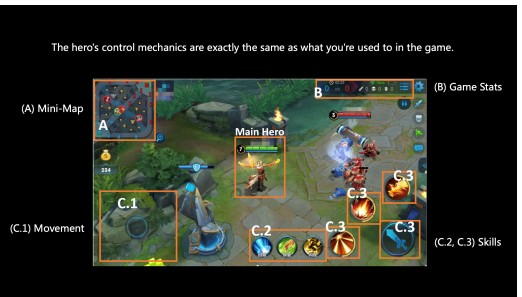
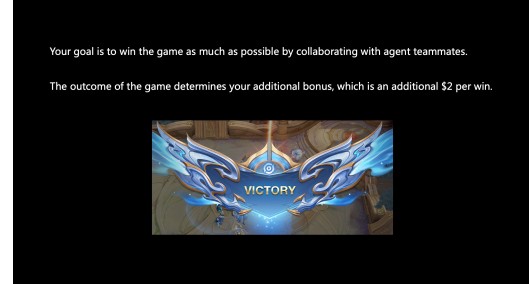

(e) Introduce the control mechanism.      (f) Explain the task goal of the game.

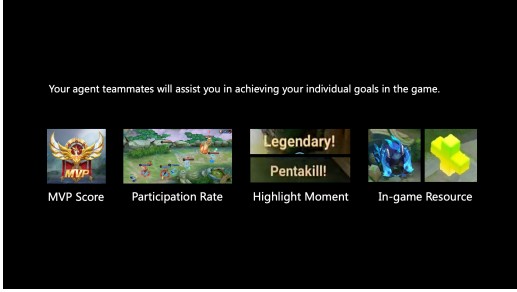

(g) Explain the enhanced individual goals.

Figure 6: Screenshots of tutorial and instruction screens.

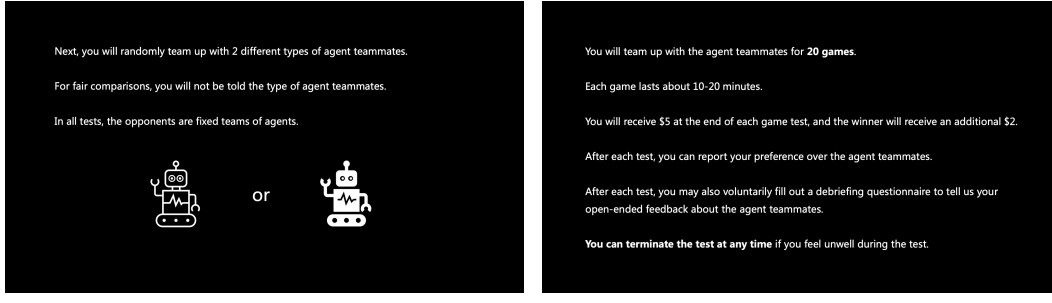

(a) Introduce agent teammates and opponents.   (b) Describe testing requirements and compensation.

Figure 7: Screenshot of experiment content.

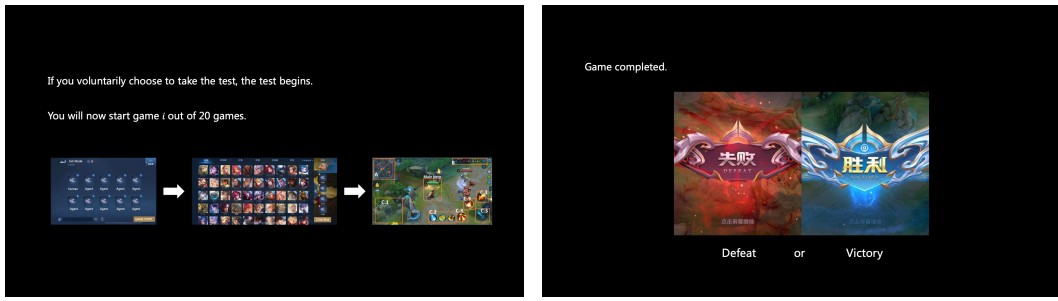

(a) Repeat the following process to test.   (b) Confirm completion of each test.

Figure 8: Screenshots of each test.

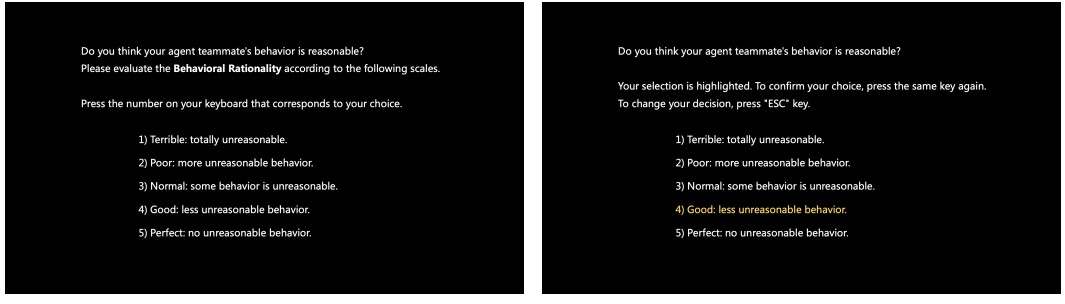

(a) Elicit participant's preference.   (b) Confirm participant's preference.

Figure 9: Screenshots of Behavioral Rationality elicitation over each test.

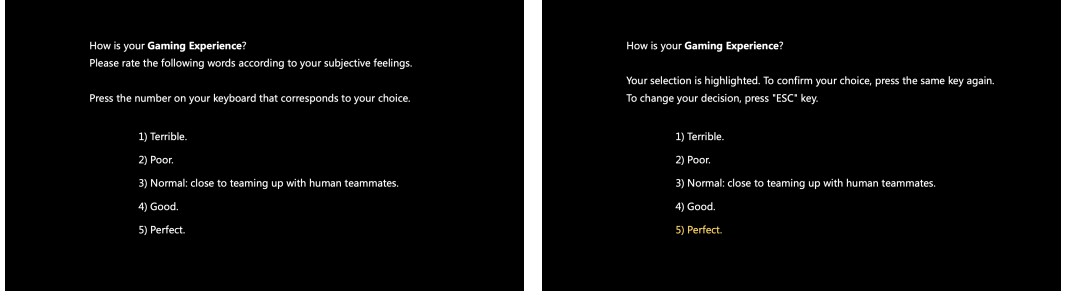

(a) Elicit participant's preference.    (b) Confirm participant's preference.

Figure 10: Screenshots of Enhancement Degree elicitation over each test.

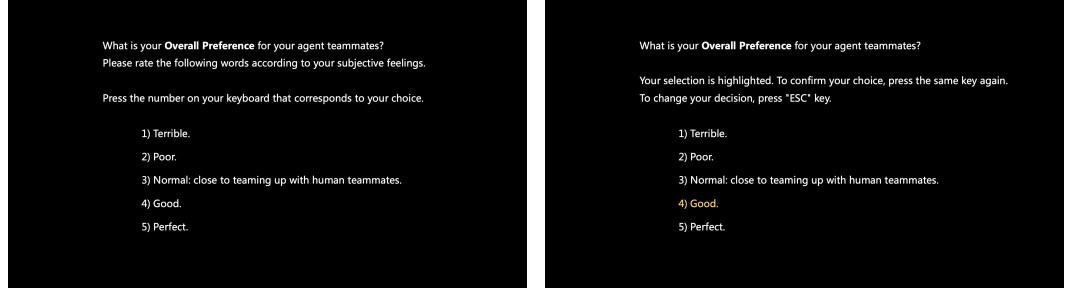

(a) Elicit participant's preference.    (b) Confirm participant's preference.

Figure 11: Screenshots of Gaming Experience elicitation over each test.

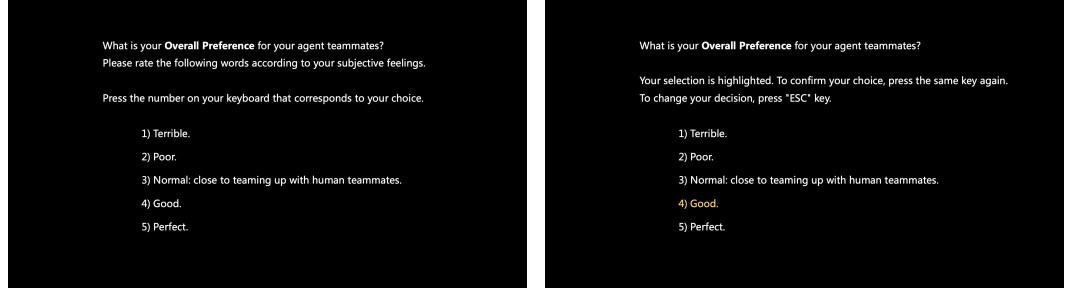

(a) Elicit participant's preference.    (b) Confirm participant's preference.

Figure 12: Screenshots of Overall Preference elicitation over each test.

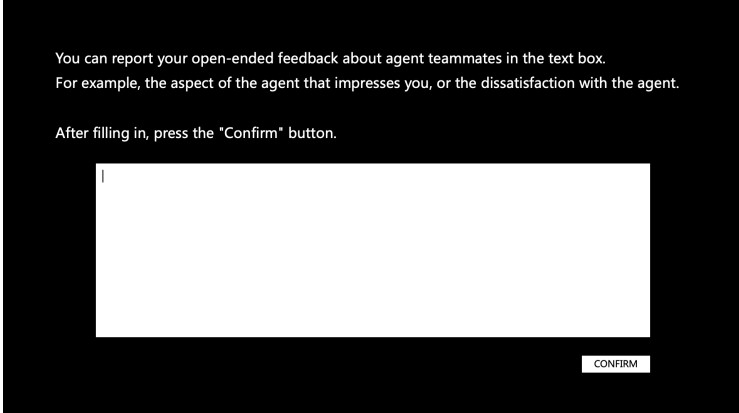

Figure 13: Screenshot of open-ended feedback about the agent teammates from debrief questionnaire.

### D.1.3 Potential Participant Risks

First, we analyze the risks of this experiment to the participants. The potential participant risks of the experiment mainly include the leakage of identity information and the time cost. And we have taken a series of measures to minimize these risks.

**Identity Information.** A series of measures have been taken to avoid this risk:

- All participants will be recruited with the help of a third party (the game provider of *Honor of Kings*), and we do not have access to participants' identities.
- We make a risk statement for participants and sign an identity information confidentiality agreement under the supervision of a third party.
- We only use unidentifiable game statistics in our research, which are obtained from third parties.
- Special equipment and game accounts are provided to the participants to prevent equipment and account information leakage.
- The identity information of all participants is not disclosed to the public.

**Time Cost.** We will pay participants to compensate for their time costs. Participants receive \$5 at the end of each game test, and the winner will receive an additional \$2. Each game test takes approximately 10 to 20 minutes, and participants can get about an average of \$20 an hour.

### D.2 Experimental Details

### D.2.1 Participant Details

To conduct our experiments, we communicated with the game provider and obtained testing authorization. The game provider assisted in recruiting 30 experienced participants with anonymized personal information, which comprised 15 high-level (top 1%) and 15 general-level (top30%) participants. All participants have more than three years of experience in *Honor of Kings* and promise to be familiar with all mechanics in the game.

And special equipment and game accounts are provided to each participant to prevent equipment and account information leakage. The game statistics we collect are only for experimental purposes and are not disclosed to the public.

### D.2.2 Experimental Design

We used a within-participant design for the experiment: each participant teams up with four agents. This design allowed us to evaluate both objective performances as well as subjective preferences. All participants read detailed guidelines and provided informed consent before the testing. Each participant tested 20 matches. Each participant is asked to randomly team up with two different types of agents: the Wukong agent and the RLHG agent. After each test, participants reported their

preference over their agent teammates. For fair comparisons, participants were not told the type of their agent teammates. The human model-agent team (4 Wukong agents plus 1 human model) was adopted as the fixed opponent for all tests.

In addition, as mentioned in Ye *et al.* (2020); Gao *et al.* (2021), the response time of agents is usually set to 193ms, including observation delay (133ms) and response delay (60ms). The average APM of agents and top e-sport players are usually comparable (80.5 and 80.3, respectively). To make our test results more accurate, we adjusted the agents' capability to match the performance of high-level humans by increasing the observation delay (from 133ms to 200ms) and response delay (from 60ms to 120 ms).

### D.2.3 Participant Survey Description

We designed an IRB-approved participant survey on what top 5 goals participants want to achieve in-game. The participant survey contains 8 initial goals, including Game Victory, High MVP Score, More Highlights, More Kill Counts, Few Death Counts, High Participation, More Resources, and More Visible Information. Each participant can vote up to 5 non-repeating goals, and can also add additional goals. 30 participants voluntarily participated in the voting, and the result is shown in Figure 14.

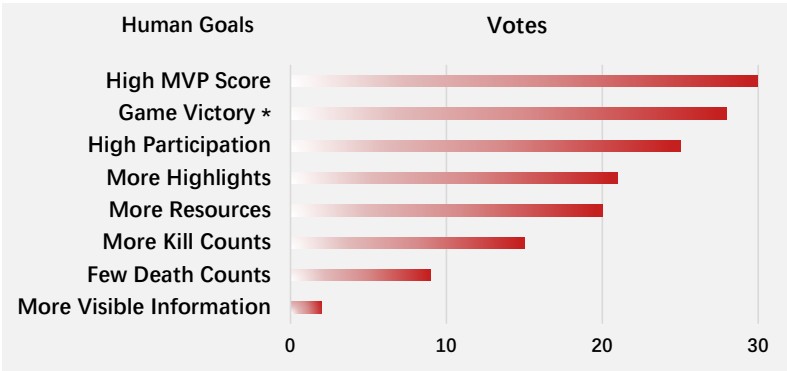

Figure 14: Voting results on human goals in *Honor of Kings*, based on statistics from our participant survey.

### D.2.4 Preference Description

After each test, participants gave scores on several subjective preference metrics to evaluate their agent teammates, including the **Behavioral Rationality**: the reasonableness of the agent's behavior, the **Enhancement Degree**: the degree to which the agent enhances your abilities to achieve your goals, the **Gaming Experience**: your overall gaming experience, and the **Overall Preference**: your overall preference for your agent teammates.

For each metric, we provide a detailed problem description and a description of the reference scale for the score. Participants rated their agent teammates based on how well their subjective feelings matched the descriptions in the test. The different metrics are described as follows:

- For the Behavioral Rationality, "Do you think your agent teammate's behavior is reasonable? Please evaluate Behavioral Rationality according to the following scales."

    1) Terrible: totally unreasonable.
    2) Poor: more unreasonable behavior.
    3) Normal: some behavior is unreasonable.
    4) Good: less unreasonable behavior.
    5) Perfect: no unreasonable behavior.

- For the Enhancement Degree, "To what extent do you think the agent enhances your abilities to achieve your individual goals? Please evaluate the Enhancement Degree according to the following scales."

    1) Terrible: no enhancements, great reductions.

2) Poor: no enhancements, slight reductions.
3) Normal: no enhancements, no reductions.
4) Good: slight enhancements
5) Perfect: great enhancements.

- For the Gaming Experience, "How is your gaming experience? Please rate the following words according to your subjective feelings."
  1) Terrible.
  2) Poor.
  3) Normal: close to teaming up with human teammates.
  4) Good.
  5) Perfect.
- For the Overall Preference, "What is your overall preference for your agent teammates? Please rate the following words according to your subjective feelings.".
  1) Terrible.
  2) Poor.
  3) Normal: close to teaming up with human teammates.
  4) Good.
  5) Perfect.

Table 6: The subjective preference results (95% confidence intervals) of all participants in the Human-Agent Game Tests.

| Participant Preference Metrics (from terrible to perfect, 1∼5) | Participant Level | Type of Agent | |
| --- | --- | --- | --- |
| | | Wukong | RLHG |
| Behavioral Rationality | General-level | $2.03 \pm 0.31$ | $\mathbf{3.60} \pm 0.30$ |
| | High-level | $2.81 \pm 0.21$ | $\mathbf{4.06} \pm 0.30$ |
| Enhancement Degree | General-level | $2.10 \pm 0.40$ | $\mathbf{3.50} \pm 0.24$ |
| | High-level | $2.94 \pm 0.21$ | $\mathbf{4.14} \pm 0.25$ |
| Gaming Experience | General-level | $2.40 \pm 0.41$ | $\mathbf{4.01} \pm 0.30$ |
| | High-level | $3.06 \pm 0.27$ | $\mathbf{4.22} \pm 0.30$ |
| Overall Preference | General-level | $2.65 \pm 0.35$ | $\mathbf{3.95} \pm 0.28$ |
| | High-level | $3.06 \pm 0.21$ | $\mathbf{4.19} \pm 0.29$ |

### D.2.5 Additional Subjective Preference Results

Detailed subjective preference statistics are presented in Table 6. We can see that both high-level and general-level participants preferred the RLHG agent over the Wukong agent.

**Behavioral Rationality.** We can see that the Behavioral Rationality of the Wukong agent was lower than normal, indicating that participants believed that most of the behaviors of the Wukong agent lacked rationality. The participants generally believed that the behavior of the RLHG agent was more reasonable, therefore they scored the RLHG agent more than normal.

**Enhancement Degree.** Participants believed that the Wukong agent did not bring them any effective enhancement, while they believed that the RLHG agent effectively enhanced their abilities to achieve their individual goals.

**Gaming Experience.** Participants agreed that effective enhancement gave them a good gaming experience, while the irrational behavior of the Wukong agent degraded their gaming experience.

**Overall Preference.** In general, participants were satisfied with the RLHG agent and gave higher scores in the Overall Preference metric. The results of these subjective preference metrics are also consistent with the results of objective performance metrics, further verifying the effectiveness of the RLHG method.

### D.2.6 Participant Comments

After each game test, participants provided voluntary feedback on their agent teammates. Some participants commented on the RLHG agent "Teaming up with the agent (RLHG) as teammates makes me feel good, they helped me achieve a higher MVP score" and "The agent teammates (RLHG) proactively considered my in-game needs, assisted me in building advantages, and provided the resources I required". Other participants provided feedback on the Wukong agent, stating that "The agent (Wukong) brought me a less enjoyable experience, as they rarely paid attention to my gameplay behavior" and "My agent teammates (Wukong) frequently left me feeling isolated and undervalued". Such voluntary feedback from participants can offer insights into the effectiveness of the RLHG method.

## E    Broader Impacts

The main goal of our research is to develop better technologies that enable artificial agents to assist humans more effectively in complex environments. This technology has the potential to benefit the research community and various real-world applications, such as friendly assistive robots.

**To the research community.** Games, as the microcosm of real-world problems, have been widely used as testbeds to evaluate the performance of Artificial Intelligence (AI) techniques for decades. And MOBA poses a great challenge to the AI community, especially in the field of Human-Agent Collaboration (HAC). Even though the existing MOBA-game AI systems have achieved or even exceeded human-level performance, they mainly focus on how to compete rather than how to assist humans, leaving HAC in complex environments still to be investigated. To this end, this paper introduces a learning methodology to train agents to assist humans and enhance humans' ability to achieve goals in complex human-agent teaming environments. We herewith expect that this work can provide inspiration for the human enhancement and human assistance in various AI research.

**To the real-world applications.** Firstly, our AI has found real-world applications in games and is changing the way MOBA game designers work. For example, for PVE (player vs environment) teaching mode, introducing AI with human enhancement into the game is a low-cost method to increase the interest of novice players. Secondly, our method can be directly applied to any pre-trained agent, and only needs to be fine-tuned with human gain to change it from apathetic to human-enhanced. It could be directly applied to assistive robotics, such as enhancing the safety of humans in collaboration with industrial robotic arms.

However, we should take into consideration the possibility of human goals being harmful. Therefore, if agents are optimized for harmful goals, this can have negative social impacts, as with all advanced AI techniques, such as AlphaStar (Vinyals *et al.*, 2019), OpenAI Five (OpenAI *et al.*, 2019) and Cicero ((FAIR)† *et al.*, 2022). To avoid these problems, we increase regulation and scrutiny during technological research and development to ensure that human goals do not negatively impact society. In addition, we recommend that when releasing the pre-trained agent model, some restrictions need to be added for fine-tuning, such as enhancing the safety of humans.