# OpenReview forum: "Make You Better: Reinforcement Learning from Human Gain"
_NeurIPS.cc/2023/Conference — Submitted to NeurIPS 2023_

### Official Review · Reviewer_HuXj · 2023-07-03

**Soundness:** 3 good
**Presentation:** 3 good
**Contribution:** 2 fair
**Rating:** 3
**Confidence:** 4

**Summary:**

The authors present a new reinforcement learning objective, dubbed Reinforcement Learning from Human Gain (RLHG), that explicitly incorporates an understanding of human performance with an intervention into the objective function. They show that training with this added component improves outcomes in a MOBA, both on the overall objective of winning and on subobjective related to satisfaction.

**Strengths:**

They present an important research problem (ML systems working as collaborators) and make a good attempt at overcoming it.
They did experiments with human participants.
The domain they trained models in is non-trivial and showing results shows good ability


**Weaknesses:**

I don't believe the main claim. The authors say that by explicitly adding their more complex models of human wants/behaviours they can get better performance, but they don't compare against a model that attempts to optimize for those things directly. A perfect RL agent that has correct values for the different objectives should be able to learn the optimal policy without using their proposed more complicated methods. They only test against a model that is trained to win, and a model that is trained on short term value optimization. How does the method compare to another model trained with similar data and objectives? The lack of comparison makes this feel like they shows that PPO and deep learning work, not that there new methods are better.

**Questions:**

"compromising AI autonomy" (line 61), what does this mean? Isn't the whole point of this method to shape the agent's actions?
Where did you get 10^20000 for the state space? (line 91)
You don't need to specify that Honor of Kings is a MOBA and that it is (renowned|complex|popular|etc.) each time you reference it
Dec-POMDP is never defined
Line 215 "We were authorized" by who?

**Limitations:**

The authors do not explain why their decomposition of the loss/training loop is _theoretically_ better than any other, they only provided some _empirical_ evidence to suggest it.
The authors only discuss the RL literature in their framing/discussion. In other domains, such as recomender systems, the issue of over optimizing for a single metric to the detriment of other objectives is a major area of research. Consider looking over some data science papers from KDD as a starting point

---

> ### Author Rebuttal · Authors · 2023-08-09
>
> Thank you for carefully reviewing our paper! We greatly appreciate your feedback on our work. We provide clarification below for your questions and concerns. If you have any further questions or comments, we will be happy to discuss them further.
>
> ----
>
> **Q1**: Clarification on Weaknesses
>
> **A1**: Please allow us to provide further clarity on the RLHG agent and the two baseline agents, i.e., the Wukong agent and the HRE agent, used in the experiment.
>
> - The Wukong agent [1] utilizes the PPO algorithm to optimize the Win Rate using environmental (original) rewards.
>
> - The HRE agent utilizes the PPO algorithm to optimize both the Win Rate and Human Performance by directly incorporating human goal rewards (i.e., rewards for achieving goals) into the optimization objective.
>
> - The RLHG agent utilizes the PPO algorithm to optimize both the Win Rate and Human Performance by incorporating the positive gains of human returns under effective enhancement versus no enhancement into the optimization objective.
>
> (1) "they don't compare against a model that attempts to optimize for those things directly"  —  Both HRE and RLHG use the same human goal reward function during the optimization process. The key difference lies in how these rewards are integrated. HRE directly incorporates human goal rewards into the optimization objective, which can lead to human-agent credit assignment issues, as evidenced by the experimental results presented in Section 4.2. The HRE agent learned many ineffective enhancement behaviors that interfered with its original objective of winning the game, ultimately resulting in a significant decrease in the Win Rate. To address these issues, one potential solution is to carefully reshape the human goal reward function. However, this approach heavily relies on domain knowledge and expertise. RLHG tackles this challenge by introducing a "baseline" (the primitive performance of humans in achieving goals without enhancement) and incorporating the positive gain (the benefit of the human performance under effective enhancement over the "baseline") into the optimization objective.
>
> (2) "A perfect RL agent that has correct values for the different objectives should be able to learn the optimal policy without using their proposed more complicated methods." —  We totally agree with your viewpoint. In fact, the RLHG method can be seen as introducing a "baseline" that filters out some incorrect values (the human returns from invalid enhancement behaviors and negative enhancement behaviors) for the optimization objective.
>
> **Q2**: "compromising AI autonomy" (line 61), what does this mean?
>
> **A2**: In our work, an autonomous agent refers to an agent whose behavior is optimized with original rewards. We optimize by finetuning the pre-trained Wukong agent (optimized for winning games) using human goal rewards, and expect the optimized agent to improve the performance of humans in achieving goals while maintaining its autonomy (Win Rate) as much as possible. In the experiment, we found that the HRE agent learned a lot of ineffective enhancement behaviors, which is manifested in the fact that the agent frequently follows humans when it is not necessary, and is very vulnerable to human behavior. These ineffective enhancement behaviors greatly damage the autonomy of the agent, leading to a significant reduction in the Win Rate. In contrast, the RLHG agent only learns from its effective enhancement behaviors, which greatly guarantees the autonomy of the agent (Only a slight drop in Win Rate) while significantly improving the performance of humans to achieve goals. We have included a supplementary description of this in the main text.
>
> **Q3**: Where did you get 10^20000 for the state space?
>
> **A3**: This data comes from [2], "As for state space, the resolution of Honor of Kings map is 130,000 by 130,000 pixels, and the diameter of each unit is 1,000. At each frame, each unit may have different status such as hit points, levels, and gold. Again, the state space is at magnitude of 10^20,000 with significant simplification."  We have included this reference in the main text.
>
> **Q4**: Dec-POMDP is never defined.
>
> **A4**: Dec-POMDP is shorthand for **Dec**entralized **P**artially **O**bservable **M**arkov **D**ecision **P**rocess. We have included this definition in the main text.
>
> **Q5**: "We were authorized" by who?
>
> **A5**: We were authorized by **Ye, et al.** to use the Wukong [1] agent and the JueWu-SL [3] agent. We have included this information in the main text.
>
> ----
>
> [1] Ye, et al. Towards playing full moba games with deep reinforcement learning. NeurIPS'2020.
>
> [2] Wu, et al. Hierarchical macro strategy model for moba game ai. AAAI'2019.
>
> [3] Ye, et al. Supervised learning achieves human-level performance in moba games: A case study of honor of kings. TNNLS'2020.

---

> > ### Comment · Reviewer_HuXj · 2023-08-18
> >
> > Thank you for responding thoroughly to my questions. Reading all the responses has answered my questions.
> >
> > Reading your response and those to the other reviewers has not changed my position. I do not believe this is a significant enough contribution. I think this result should be published at another venue if the other minor issues are fixed, in particular increasing the clarity of the writing.

---

### Official Review · Reviewer_5Pqf · 2023-07-06

**Soundness:** 2 fair
**Presentation:** 1 poor
**Contribution:** 2 fair
**Rating:** 4
**Confidence:** 4

**Summary:**

This paper proposes a new method called Reinforcement Learning from Human Gain (RLHG) to effectively enhance human goal-achievement abilities in collaborative tasks with known human goals. The paper evaluates the RLHG agent in the widely popular Multi-player Online Battle Arena (MOBA) game, Honor of Kings, by conducting experiments in both simulated environments and real-world human-agent tests.

**Strengths:**

The problem setting considered by the paper is tightly connected to some real-world problems (e.g., assistive agents in MOBA games).
Experiments are performed in a real-world application (Honor of King).

**Weaknesses:**

It is difficult to evaluate the major contribution of the paper (the two-step training process) because
1) the major contribution of the paper, as the authors claimed in the paper, is orthogonal to many of the complications in the paper (e.g., multiple agents, multiple goals, partial observability). These complications are not contributions and they make it hard to understand the contribution of the paper clearly. Maybe the authors added them because their experiments are in Honor of Kings?
2) probably because the paper focuses too much on these complications, the paper fails to explain why RLHG is a good idea and provides clear evidence. For example, why do the authors propose estimating primitive human performance rather than primitive human+agent performance? What is making estimating primitive human performance helpful? Can this idea be used in environments without humans?
3) the writing of the paper is unsatisfying. I was completely lost when reading the paper. Please see Questions for my questions about the paper.
4) in experiments, the new method achieved a worse winning rate compared with the baseline method. I can understand this performance drop given that there are improvements regarding other metrics. What should I learn from this indeterminate result?

**Questions:**

Section 1:
"rather than replacing them outright" why would you talk about this? How is this related to the previous text?
"The RLHG method can be seen as a plug-in that can be directly utilized to fine-tune any pre-trained agent to be assistive in human enhancement." Do you literally mean ANY pre-trained agent?
"human goals". "human goals" sounds too broad and vague. What do you actually mean here?
"AI autonomy". ???
"such as human-agent credit assignment issues, i.e., human rewards for achieving goals are assigned to non-assisting agents, which potentially leads the agent to learn poor behaviors and forfeits its autonomy." Why is this an issue? In RL you can have environments that emit rewards even if the agent does nothing.

Section 2:
Could you explain how the agents interact with the human?
Do you consider one human?
Do you consider a finite set of global states, actions, and observations?
How are global states used?
Do you assume that the policy only depends on the current observation?
What do ''scenarios'' mean?
pi_\theta is not defined, theta is not defined.
What do R_t and R^H_t mean?
What do Eπθ [G], and EπH [G] mean?
It is important to distinguish between definition and equality. In this paper, "=" is used for both purposes.
"The agent’s policy gradient can be formulated as". I know what "policy gradient" means in classic RL. But to be honest I am not sure what "policy gradient" means here. How do you derive the following equation?


Section 3:
What does Equation 2 mean? What does "the probability of exploring each policy" mean?
Why do you separate human rewards from the agent's rewards, given that you are going to combine them together eventually?

---

> ### Author Rebuttal · Authors · 2023-08-09
>
> Thank you for carefully reviewing our paper! We greatly appreciate your feedback on our work. We provide clarification below for your questions and concerns. If you have any further questions or comments, we will be happy to discuss them further.
>
> ----
>
> **Q1**: Clarification on Contributions
>
> **A1**: We have reorganized our contributions to enhance their clarity.
> - We explored approaches to enable (surpass) human-level agents to assist humans in achieving their goals in complex collaborative environments.
> - Through this exploration, we gained insights into challenges like human-agent credit assignment. This challenge can cause agents to learn many ineffective enhancement behaviors that hinder their original goal of winning the game.
> - To address this challenge, we proposed the RLHG method and provide a detailed implementation framework.
> - We conducted human-agent tests on the popular MOBA game Honor of Kings. Both objective metrics and subjective preference results of the participants verified the effectiveness of our proposed method.
>
> **Q2**: Clarification on AI Autonomy and Human-Agent Credit Assignment
>
> **A2**: In our work, an autonomous agent refers to the pre-trained Wukong agent which is optimized for winning games using environmental (original) rewards. We aim to improve human performance in achieving goals while maintaining the agent's autonomy (Win Rate). However, during the experiment, we found that the HRE agent struggled with credit assignment when collaborating with humans. This led to ineffective enhancement behaviors, such as following humans unnecessarily (even in scenarios where humans can accomplish certain goals independently) or susceptibility to human behaviors. These behaviors significantly impeded the agent's autonomy, resulting in a significant decrease in Win Rate. To address this, one potential solution is to carefully reshape the human goal reward function. However, this approach heavily relies on domain knowledge and expertise. RLHG solves this by introducing a "baseline" (the primitive performance of humans in achieving goals without enhancement) and incorporating the positive gain (the benefit of the human performance under effective enhancement over the "baseline") into the optimization objective.
>
> **Q3**:  Can this idea be used in environments without humans?
>
> **A3**: The RLHG framework can be naturally extended by replacing the human model with other specified surrogate models. However, our research primarily concentrates on human enhancement, which holds greater practical significance.
>
> **Q4**：Clarification on Win Rate and Human Metrics
>
> **A4**: Firstly, we would like to clarify that the pre-trained Wukong agent used in our experiment is beyond the human level. Secondly, the Wukong agent is trained for winning the game using environmental rewards. In contrast, the RLHG agent introduced an additional optimization objective of enhancing human performance. However, there is a natural trade-off between Win Rate and Human Metrics. Thirdly, our participant survey results show that many participants were more concerned with achieving their individual goals rather than the game result. Our human-agent test results also show that many participants preferred teaming with the RLHG agent, despite its slightly lower Win Rate compared to teaming with the Wukong agent. Finally, we also conducted an ablation study on the balance parameter $\alpha$ in Appendix C.1 and proposed an adaptive adjustment mechanism for better practical application in Appendix C.2.
>
> **Q5&A5**: For Section 1
> - "rather than replacing them outright" -> It is related to Figure 1 (main text), in which the human wants the coin. At this point, the agent should act as an assistant to help humans pass through, so that the human can obtain the coin, rather than replacing the human to obtain the coin, which would prevent the human from achieving his/her goal.
> - "Do you literally mean ANY pre-trained agent?" -> Although the RLHG framework does not impose constraints on pre-trained agents, its practical utility may be limited for low-level agents which may not possess the innate ability to assist human players.
> - "Human goals" -> In our work, it refers to the individual goals that humans want to achieve during the task.
>
> **Q6&A6**: For Section 2
> - "how the agents interact with the human?" -> Both Figure 7 (main text) and the video (line 201) demonstrate the interaction between the agents and the human.
> - "Do you consider a finite set of global states, actions, and observations?" -> Please refer to the detailed description of the state and action spaces in Appendix B.
> - "Do you assume that the policy only depends on the current observation?" -> In Section 3.3, we stated that we used an LSTM module (step size is 16) to process the observed historical information.
> - "Notation definition" -> We have clarified the notations in the paper to improve its readability, including: $\pi_\theta$ represents the agent's policy network and $\theta$ is the network parameter. $R_t$ and $R^H_t$ denote the original reward and human goal reward at timestep $t$, respectively. $E_{\pi_\theta}[G]$ and $E_{\pi^H}[G]$ represent the expected return based on $\pi_\theta$ and $\pi^H$, respectively.
> - "Policy gradient" -> We have clarified the agent's policy gradient as:
> > $g(\theta)=Eo_{0:\infty},a_{0:\infty}[\sum_{t=0}^{\infty}Ea_t^H\sim\pi^H[ A_t + A^H_t ]\nabla_{\theta}log\pi_{\theta}(a_t|o_t,a_t^H)]$
>
> **Q7&A7**: For Section 3
> - "Equation 2" -> It expresses that we consider $\pi_\theta$ to be $\pi_{\theta}^{ef}$, $\pi_{\theta}^{in}$, or $\pi_{\theta}^{ne}$, depending on whether the human value function $V_H^{\pi_{\theta}, \pi^H}(s)$ based to the policy $\pi_\theta$ is greater than, equal to, or less than the human primitive value $V_H^{\pi, \pi^H}(s)$.
> - "Why separate rewards" -> Separating them facilitates a clearer description of each, as our approach primarily focuses on improving the estimation of human enhancement advantages.

---

> > ### Comment · Area_Chair_2WRg · 2023-08-19
> >
> > Dear authors,
> >
> > thank you for submitting your response to the comments.
> >
> > Dear Reviewer 5Pqf,
> >
> > were your concerns addressed by the authors?
> >
> > Best,
> >
> > AC

---

### Official Review · Reviewer_3jnU · 2023-07-06

**Soundness:** 3 good
**Presentation:** 3 good
**Contribution:** 3 good
**Rating:** 6
**Confidence:** 4

**Summary:**

The authors propose Reinforcement Learning from Human Gain, an RL algorithm that explicitly optimizes for enhancing human abilities in cooperative human-AI settings. Given a predefined set of human goals, the main approach first learns a value network to estimate the primitive human performance at achieving said goals. Then, a secondary Gain network is trained to estimate the enhancement the human return receives under interactions with the cooperative agent. The cooperative policy is trained with a combination of a traditional agent value network and the proposed human gain-based value network. The authors test the RLHG framework in a cooperative game and find that across experiments with real humans, the RLHG agent is preferred over an agent without the human gain objective, despite having a lower overall win rate.

**Strengths:**

- The problem setting is interesting and relevant, and the proposed solution of optimizing for human gain is intuitive — it makes sense that a cooperative agent should account for improvements in human behaviours, rather than having the agent directly optimize for its own reward or what it perceives human rewards to be (which may reduce human enjoyment and overall autonomy in a task, as noted by the authors).
- The experiments use a complex multiagent task and test both human models and real human participants. The results show a significant improvement in human preference for the RLHG agent across the predefined goals as well as various subjective metrics.

**Weaknesses:**

- The method seems sensitive to the choice of partner $\pi$ while collecting the primitive human episodes. If the initial partner is already very good, will its success be attributed to the human? And vice versa -- if the partner is very bad, would that lead to a false representation of the base human skill? The paper would be strengthened with additional studies on how sensitive the method is to different pretrained policies.
- The method relies on knowing the set of human goals beforehand. It would be interesting to have additional analyses of how the method is affected in the case where some of the goals are missing or misspecified, which is more representative of a real-world scenario.

**Questions:**

What about scenarios where we don’t know human goals beforehand? This method relies on being able to accurately measure human rewards/goals which may not always be feasible (ill-defined goal spaces, human preferences changing over time). The authors comment on combining RLHG with goal inference methods, but those methods may still be faulty or difficult to learn. Could this method be applied to more general metrics of 'gain'-- e.g. increasing empowerment in the general assistance case?

**Limitations:**

The authors adequately discuss the limitations of this work. Additional discussion on societal impacts would be helpful, as these agents are trained to interact with people directly.

---

> ### Author Rebuttal · Authors · 2023-08-09
>
> Thank you for carefully reviewing our paper! We greatly appreciate your positive feedback on our work. We provide clarification below for your questions and concerns. If you have any further questions or comments, we will be happy to discuss them further.
>
> ----
>
> **Q1**: how sensitive the method is to different pretrained policies.
>
> **A1**:  Our research is motivated by the desire to enable (surpass) human-level agents to assist humans in achieving their goals in complex collaborative environments. Although the RLHG framework does not impose constraints on pre-trained agents, its practical utility may be limited for low-level agents, as these agents may not possess the innate ability to assist human players. For high-level agents (such as the Wukong agent), the human-agent test results show that both objective metrics and subjective preferences of participants teaming with RLHG agents are better than those of teaming with Wukong agents, which verifies the effectiveness of the RLHG method.
>
> **Q2**: Could this method be applied scenarios where we don’t know human goals?
>
> **A2**: The experimental results presented in Section 4 indicate that the RLHG method can effectively enhance human performance in achieving goals in scenarios where human goals are known. In more general scenarios where human goals are not directly accessible (e.g., when they are implicitly included in human states and trajectories),  combining the RLHG method with empowerment-based approaches like AvE [1] is promising. Specifically, the diversity of final states (human performance related) can serve as a proxy for measuring human empowerment capacity. This proxy can be viewed as an intrinsic "human reward" and can be directly integrated into the RLHG framework. To implement this approach, the RLHG method initially trains a value network to estimate primitive human empowerment using human-agent team play trajectories. Subsequently, the RLHG method trains a gain network that aims to effectively improve human empowerment.
>
> **Q3**: Lack of discussion on societal impacts.
>
> **A3**: We discussed the societal impacts of our work, including its implications for the AI research community and real-world applications, in Section E of the Appendix. We have now moved this discussion to the main text to underscore its significance.
>
> ----
>
> [1] Du, et al. AvE: Assistance via empowerment. NeurIPS'2020.

---

> > ### Comment · Reviewer_3jnU · 2023-08-14
> >
> > Thank you for the response and clarifications. I think it would be helpful to add some of the answers above to the draft (impact of quality of pretrained agent, etc.). I will be keeping my current score.

---

### Official Review · Reviewer_11JW · 2023-07-06

**Soundness:** 3 good
**Presentation:** 3 good
**Contribution:** 2 fair
**Rating:** 6
**Confidence:** 3

**Summary:**

This paper focuses on the fine-tuning of a pre-trained agent to assist and enhance the performance of a given human model in achieving specific goals. The authors assume access to a human model and a pre-trained agent. The authors propose a two-step approach.

1. The human model's initial performance is determined by training a value network to estimate its effectiveness in goal attainment using episodes generated through joint execution with the agent.

2. The is trained agent to learn effective behaviors for enhancing the human model's performance using a gain network that estimates the improvement in human return when compared to the initial performance.

The algorithm is evaluated on a Multi-player Online Battle Arena (MOBA) game.

**Strengths:**

The idea of developing algorithms to assist humans to solve tasks is interesting and of great practical interest, and this paper makes headway in this direction. The authors conduct extensive experiments which includes evaluating using real players to test their algorithm in a game.

**Weaknesses:**


1. The algorithmic contribution in this paper appears to be relatively modest, as it mainly builds upon the existing Proximal Policy Optimization (PPO) approach. The authors introduce a gain function, which essentially computes an advantage by comparing it to another state-dependent baseline ( $V_\phi(s)$)

2. The assumption of having a human model is justifiable; however, the strong reliance on assuming knowledge of human goals, in my opinion, limits the direct applicability of this research (as acknowledged in the limitations section).

3. There is a lot of notational ambiguity in the paper (Section 2.2), which makes reading a little hard. For example,

3a. Advantage function generally depends on state/observation and action. In this setting, the Advantage is independent of both.

3b. Is V value of a state or the infinite horizon discounted reward? It is unclear as its used in both contexts.

3c. G is used as return-to-go, which should be a state dependent function.

4. I do not understand the exact need for the gain network. What would happen if in line 12 of the algorithm, you drop the - Gain(s) part? This essentially means that you are computing advantage with respect to the human primitive baseline.

5. The authors have invested significant effort and computational resources in conducting their experiments, making it extremely challenging to reproduce or recreate such experiments due to the demanding compute requirements. Although this does not diminish the value of their work, it would be beneficial if the authors could incorporate simpler environments into their experimental setup. This addition would aid in evaluating the algorithm's performance and further validate its quality.

**Questions:**

See weaknesses.

**Limitations:**

The authors discuss limitations and future work.

---

> ### Author Rebuttal · Authors · 2023-08-09
>
> Thank you for carefully reviewing our paper!  We greatly appreciate your positive feedback on our work. We provide clarifications below for your questions and concerns. If you have any further questions or comments, we will be happy to discuss them further.
>
> ----
>
> **Q1**: Clarification on Contributions.
>
> **A1**: The RLHG algorithm is a key component of our contribution, but our work extends beyond this. Specifically, we have made the following contributions:
>
> - We explored approaches to enable (surpass) human-level agents to assist humans in achieving their goals in complex collaborative environments.
>
> - Through this exploration, we gained insights into problems like human-agent credit assignment issues. These issues can cause agents to learn many ineffective enhancement behaviors that hinder their original goal of winning the game.
>
> - To address this challenge, we propose the RLHG method and provide a detailed implementation framework.
>
> - We conducted human-agent tests on the popular MOBA game Honor of Kings. Both objective metrics and subjective preferences of the participants verified the effectiveness of our proposed method.
>
> **Q2**: Clarification on Known Human Goals Setting.
>
> **A2**: The setting of our research work is in scenarios where human goals are known, in complex collaborative environments. In more general scenarios where human goals are not directly accessible, combining the RLHG method with goal inference approaches like Bayesian Delegation [1] and empowerment-based approaches like Assistance via Empowerment (AvE) [2] is promising.  For example, consider combining the RLHG method with the AvE method. The diversity of final states (human performance related) can serve as a proxy for measuring human empowerment capacity. This proxy can be viewed as an intrinsic "human reward" and can be directly integrated into the RLHG framework.
>
> **Q3**: Clarification on Notational Ambiguity.
>
> **A3**: We have clarified and disambiguated the notations used in the paper to improve its readability and comprehension, including but not limited to:
>
> - Advantage function $A$ -> $A(s_t,a_t)=Q(s_t,a_t)-V(s_t)$
>
> - Value function $V$ -> $V(s_t)=E[G_t|s_t]$
>
> - Return $G$ -> $G_t=\sum_{k=0}^\infty \gamma^k R_{t+k+1}$
>
> **Q4**: Clarification on Gain Network.
>
> **A4**: The gain network $Gain_\omega(s)$ is a value network that estimates the expected benefit of human returns $Gain^{\pi_\theta^{ef}, \pi^H}$ resulting from effective enhancement versus no enhancement (human primitive value) for a given state $s$. We refer to this benefit as **Human Positive Gain**. The role of the gain network $Gain_\omega(s)$ in line 12 of the algorithm can be interpreted from two perspectives:
>
> (1)  Advantage of return over effective enhancement value. In state $s$, it measures how much better taking a specific action $a$ compared to the expected value $V_\phi(s)+Gain_\omega(s)$ of taking any effective enhancement action.
>
> (2)  Advantage of gain over expected positive gain. In state $s$, it measures how much better taking a specific action $a$ compared to the gain of taking any action that can produce a positive gain.
>
> By $- Gain_\omega(s)$, the agent can be guided to take actions with higher potential for human returns, rather than merely actions that exceed the human primitive value.
>
> **Q5**: Clarification on Experimental Environment.
>
> **A5**: Thank you for your suggestions. Our research is motivated by the desire to enable high-level agents to assist humans in achieving their goals in complex collaborative environments. In future work, we plan to apply the RLHG framework to games such as Google Football and StarCraft, to facilitate easier reproduction of our method by other researchers as well.
>
> ----
>
> [1] Sarah, et al. Too many cooks: Bayesian inference for coordinating multi-agent collaboration. TOPICS'2021.
>
> [2] Du, et al. AvE: Assistance via empowerment. NeurIPS'2020.

---

> > ### Comment · Reviewer_11JW · 2023-08-10
> >
> > Thank you for your and response and the change in notation. I think it will improves the readability of the paper.  I retain my positive endorsement of the work.

---

### Author Rebuttal · Authors · 2023-08-09

Thanks to all reviewers for carefully reviewing our paper! We are grateful for your valuable feedback and suggestions, which we have addressed and incorporated into the revised manuscript. If you have any further questions or comments, we would be more than happy to discuss them.

Moreover, we have included the experimental findings of Appendix C.1 (Ablation study on the balance parameter $\alpha$ for Win Rate and Human Performance) and Appendix C.2 (Experiments on the adaptive adjustment mechanism) in the supplementary PDF for your reference, if necessary.

---

### Decision · Program_Chairs · 2023-09-21

**Decision:**

Reject

**Comment:**

This paper introduces RLHG which aims to enhance human performance in achieving specific goals through collaboration with AI agents. The authors propose a two-step approach: first, a value network is trained to assess the human model's initial effectiveness in goal attainment when working with the agent. Then, Gain network is trained to estimate the improvement in human performance when interacting with the cooperative agent. To train the cooperative policy, a hybrid approach is adopted, combining a conventional agent's value network with the newly introduced human gain-based value network.

Even though reviewers and AC think that the problem setting is interesting, I am recommending reject due to the following concerns:

* Writing quality needs to be improved. Several reviewers have raised concerns about unclear motivation/contribution due to this.
* As Reviewer 3jnU, analysis on sensitive to the choice of partner would be interesting
* Some assumptions (e.g., knowing the set of human goals) are restricting the generality of the proposed method